# A regulatory module mediating temperature control of cell-cell communication facilitates tree bud dormancy release

Shashank K Pandey [1,10], Jay Prakash Maurya [1,2,10], Bibek Aryal[1,11], Kamil Drynda [3,11], Aswin Nair[1], Pal Miskolczi[1], Rajesh Kumar Singh [1,4], Xiaobin Wang[1,5], Yujiao Ma [1,6], Tatiana de Souza Moraes[7], Emmanuelle M Bayer [7], Etienne Farcot[3], George W Bassel[8], Leah R Band[3,9] & Rishikesh P Bhalerao [1✉]

## Abstract

**The control of cell–cell communication via plasmodesmata (PD) plays a key role in plant development. In tree buds, low-temperature conditions (LT) induce a switch in plasmodesmata from a closed to an open state, which restores cell-to-cell communication in the shoot apex and releases dormancy. Using genetic and cell-biological approaches, we have identified a previously uncharacterized transcription factor, Low-temperature-Induced MADS-box 1 (LIM1), as an LT-induced, direct upstream activator of the gibberellic acid (GA) pathway. The LIM1-GA module mediates low temperature-induced plasmodesmata opening, by negatively regulating callose accumulation to promote dormancy release. LIM1 also activates expression of FT1 (FLOWERING LOCUS T), another LT-induced factor, with LIM1-FT1 forming a coherent feedforward loop converging on low-temperature regulation of gibberellin signaling in dormancy release. Mathematical modeling and experimental validation suggest that negative feedback regulation of LIM1 by gibberellin could play a crucial role in maintaining the robust temporal regulation of bud responses to low temperature. These results reveal genetic factors linking temperature control of cell–cell communication with regulation of seasonally-aligned growth crucial for adaptation of trees.**

**Keywords** Dormancy; Temperature; Plasmodesmata; Callose; Gibberellins
**Subject Category** Plant Biology

## Introduction

The synchronization of plant growth with seasonal changes is crucial for the survival of perennial trees in boreal and temperate regions. Photoperiod and temperature are the two main environmental cues regulating environmental control of seasonal growth (Yamane et al, 2023; Yang et al, 2021; Zhang et al, 2023). Signaling the advent of winter, the reduction in day length (Nitsch, 1957; Vince-Prue, 1975) and temperature (Heide and Prestrud, 2005; Svystun et al, 2019) suppresses the formation of new leaf primordia and shoot apical meristem (SAM) activity. The arrested leaf primordia and SAM are enclosed within an apical bud (Ruttink et al, 2007; Singh et al, 2017). Growth arrest of the bud is then maintained by dormancy. After a prolonged exposure to low temperatures, dormancy is released and growth resumes in the spring as temperature rises (Singh et al, 2017; Weiser, 1970).

The environmental control of cell–cell communication via plasmodesmata (PD) plays a crucial role in regulating seasonal growth cycles in trees. PD are channels that facilitate cell–cell movement of developmental regulators such as proteins, short RNAs, and hormones (Li et al, 2021; Maule et al, 2011). Short photoperiods induce the accumulation of electron-dense callose-rich dormancy sphincters, which contribute to the closure of PD (Jian and Sun, 1992; Rinne and Schoot, 1998; Tylewicz et al, 2018). As a result, PD-mediated intercellular communication is suppressed in the SAM, making it insensitive to growth-promoting signals and thereby establishing dormancy in apical buds. The mechanism inducing PD blockage by photoperiodic signal is well understood. In hybrid aspen as well as birch and peach, the plant hormone ABA acting via a MADS-box transcription factor *SHORT VEGETATIVE PHASE LIKE* (*SVL*) and related *DAM* genes (homologs of *SVP*, a floral repressor in *Arabidopsis*), play a crucial role in PD closure downstream of short days (SDs) (Singh et al, 2019; Tylewicz et al, 2018; Zhao et al, 2023). This is a conserved

[1]Umeå Plant Science Centre, Department of Forest Genetics and Plant Physiology, Swedish University of Agricultural Sciences, SE-901 87 Umeå, Sweden. [2]Plant Development and Molecular Biology Lab, Department of Botany, Institute of Science, Banaras Hindu University, Varanasi 221005 Uttar Pradesh, India. [3]Centre for Mathematical Medicine and Biology, School of Mathematical Sciences, University of Nottingham, Nottingham NG7 2RD, UK. [4]Biotechnology Division, CSIR-Institute of Himalayan Bioresource Technology, Palampur, Himachal Pradesh 176061, India. [5]College of Agriculture and Biotechnology, Zhejiang University, Hangzhou, Zhejiang 310058, P. R. China. [6]Shandong Academy of Grape, Jinan, Shandong 250100, P. R. China. [7]Laboratoire de Biogenèse Membranaire, UMR5200, CNRS, Université de Bordeaux, Villenave d'Ornon, France. [8]School of Life Sciences, University of Warwick, Coventry CV4 7AL, UK. [9]Division of Plant and Crop Sciences, School of Biosciences, University of Nottingham, Sutton Bonington Campus, Loughborough LE12 5RD, UK. [10]These authors contributed equally as first authors: Shashank K Pandey, Jay Prakash Maurya. [11]These authors contributed equally as second authors: Bibek Aryal, Kamil Drynda. ✉E-mail: Rishi.Bhalerao@slu.se

pathway, as shown by the role played by ABA-inducible ABF3 transcription factor in dormancy in the pear (Yang et al, 2023), as well as closure of PD induced by ABA in the lower plant *Physcomitrella patens* (Kitagawa et al, 2019).

Conversely to closure of PDs during dormancy establishment, low-temperature conditions (LT) induce dormancy release, which is associated with the removal of dormancy sphincters that presumably lead to the opening of PD (Rinne et al, 2011). In contrast with SD induced PD closure, the mechanism underlying PD opening and its relationship with dormancy release by LT is not as well understood. For example, while exogenous application of GA to axillary buds of hybrid aspen induces loss of dormancy sphincters (Rinne et al, 2011), endogenous role of GA in PD opening is not well established. Intriguingly, hybrid aspen mutants of *FLOWERING LOCUS T 1* (*FT1*) are severely compromised in bud break even after exposure to LT (suggesting defects in dormancy release) (André et al, 2022; Sheng et al, 2023) and yet dormancy sphincters are removed from PD in the *ft1* mutant buds (André et al, 2022). Also, the upstream regulators that mediate LT activation of GA and *FT1* implicated in PD and dormancy regulation are not well characterized. Consequently, how LT induces PD opening, a key feature of the dormancy release, is poorly understood, and our understanding of the LT-controlled dormancy release mechanism remains incomplete. To help address this knowledge gap, using hybrid aspen as a model, we have identified the key role of *LIM1* (*Low-temperature Induced MADS-box 1*) a previously uncharacterized MADS-box transcription factor. We show that plant hormone GA and *FT1* as downstream targets of *LIM1* and that LIM1-GA module suppresses callose accumulation to promote PD opening in response to LT. Thus, LT acts antagonistically to short days, converging on *LIM1* to induce its expression, and the interplay of *LIM1* with GA mediates the environmental control of PD dynamics in the shoot apex to regulate dormancy and bud break.

## Results

### LT activates the expression of *LIM1* in dormant buds

To determine the genetic regulators of LT-induced PD opening, we screened gene expression data for transcription factors that were induced in hybrid aspen dormant buds after LT, correlating with opening of PD (Karlberg et al, 2010) and selected transcription factors whose expression matched LT induction of *FT1* and *GA20-oxidase* (a key enzyme in GA biosynthetic pathway), (André et al, 2022; Eriksson et al, 2000; Sheng et al, 2023), the key components implicated in dormancy release and bud break (Singh et al, 2018). This analysis revealed 88 transcription factors induced by LT. Of these LT-induced transcription factors, was a previously uncharacterized MADS-box transcription factor, *LIM1* (Appendix Fig. S1). *LIM1* encodes a protein with 208 amino acid and has conserved MADS-box and K-box, characteristic of MADS-box transcription factors (Appendix Fig. S1A). Transcription factors of MADS-box family have been implicated in dormancy induction, in several tree species (Moser et al, 2020; Singh et al, 2019; Yamane et al, 2019; Zhao et al, 2023). In contrast, antagonistically acting, promoters of dormancy release are much less studied. Moreover, a large number of genes in *Populus* are often encoded by two closely related

paralogs due to genome duplication (Tuskan et al, 2006), whereas *LIM1* lacks such a closely related paralog (Appendix Fig. S1B). Thus, based on these criteria, we selected *LIM1* for further functional analysis.

We then performed a detailed analysis of *LIM1* expression in the SD-induced dormant buds of wild-type (WT) hybrid aspen plants, before and after LT treatments to induce dormancy release, and subsequently during bud break following transfer to warmer temperatures (Fig. 1). A previously characterized LT induced gene *EBB3* (Azeez et al, 2021) was used as a positive control and *Potrx047715g14206*, a gene whose expression is not changed by LT was used a negative control (Appendix Fig. S2). Compared with low levels in dormant buds (after 11 weeks of SD; 11WSD), *LIM1* expression was progressively upregulated after 2 and 5 weeks of LT (2WC and 5WC, respectively). Following transfer to warmer temperatures after dormancy release to induce bud break (2 weeks of long days; 2WLD), *LIM1* expression was downregulated compared with that after 5WC (Fig. 1). However, despite this downregulation, *LIM1* expression was still much higher during bud break compared with dormant buds (11WSD). This suggests that exposure to LT induces *LIM1* expression from low levels in dormant buds to progressively higher levels, and this induction scales with increasing exposure to LT, correlating with dormancy release.

### *SVL* and *FT1* independent activation of *LIM1* by LT

We next investigated the pathways mediating LT activation of *LIM1*. *DAM/SVL* genes promote dormancy, and LT downregulates their expression during dormancy release in several plants, including hybrid aspen (da Silveira Falavigna et al, 2021; Sasaki et al, 2011; Singh et al, 2018; Singh et al, 2019; Wu et al, 2017; Zhao et al, 2023). We investigated whether *SVL* is a repressor of *LIM1* expression and, if so, whether LT induces *LIM1* expression by

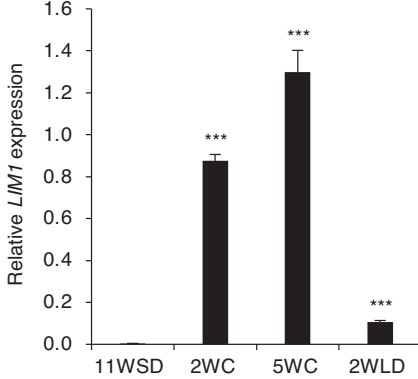

**Figure 1. Low temperatures activate the expression of *LIM1*.**

The expression pattern of *LIM1* in wild-type buds after 11 weeks of short-day (SD), followed by 2 and 5 weeks of cold temperature (2WC, 5WC) and after 2 weeks subsequent exposure to long days and warmer temperatures (2WLD). Expression values shown are normalized to the reference gene UBQ and are averages of three biological replicates. Error bars indicate standard error mean (±SEM). Statistical analysis was done using unpaired t-test. Asterisks (***) indicate significant difference (***$P < 0.001$) with respect to 11WSD. $P$ values $= 1.88 \times 10^{-5}$ (11WSD vs 2WC), 0.0005 (11WSD vs 5WC), and 0.00022 (11WSD vs 2WLD). Source data are available online for this figure.

suppression of *SVL*. *LIM1* expression in LT-treated buds of *SVL*-downregulated plants (SVL-RNAi) was compared with WT control buds, revealing no significant difference (Appendix Fig. S3A). Thus, *LIM1* upregulation by LT is independent of *SVL* suppression.

Hybrid aspen *FT1* is also induced by LT and plays a crucial role in the LT response of buds, which is essential for bud break (André et al, 2022). We therefore also investigated whether LT upregulation of *LIM1* is dependent on *FT1*, by comparing *LIM1* expression in LT-treated *ft1* mutant buds. LT was able to induce *LIM1* expression to the same levels in buds of the *ft1* mutant as the WT control (Appendix Fig. S3B), suggesting that *LIM1* induction by LT is independent of LT regulation of *FT1*. Altogether these results suggest that LIM1 is not a downstream target of *SVL* or *FT1* pathway in LT response and distinct pathways may be involved in regulating the repression of *SVL* and induction of *LIM1* by LT.

## LIM1 mediates dormancy release and promotes bud break

To investigate *LIM1*'s role in dormancy and bud break, we generated transgenic hybrid aspen plants overexpressing *LIM1* (LIM1oe) (Appendix Fig. S4A), as well as plants with reduced *LIM1* expression (LIM1-RNAi) (Appendix Fig. S4B). We then exposed WT, LIM1oe, and LIM1-RNAi plants to 11 weeks of SDs (11WSD), followed by exposure to LT to induce dormancy release, and then LD to induce bud break. The LIM1oe plants presented significantly earlier bud break compared with WT (Fig. 2A,D). Conversely, LIM1-RNAi plants showed a significant delay in bud break compared with WT (Fig. 2B,E). These results indicate that LIM1 acts to promote bud break.

Interestingly, in contrast with wild type, bud break could be initiated in LIM1oe plants even if plants were transferred directly to LD after 11WSD, without prior exposure to LT that is otherwise essential for dormancy release (Fig. 2C). This suggests that LIM1 negatively regulates dormancy and hence enhancing *LIM1* expression could bypass the requirement of LT for dormancy release. These results together with the induction of *LIM1* expression in buds by LT, coinciding with dormancy release, would be consistent with *LIM1* function in mediating LT-induced dormancy release in the buds and subsequent bud break, post dormancy release.

## LIM1 suppresses callose accumulation and mediates PD opening

Closure of PD is essential for dormancy in buds (Tylewicz et al, 2018). Conversely, LT induces dormancy release, which is associated with the opening of PD (Rinne et al, 2011; Rinne and Schoot, 1998). As PD opening and dormancy release correlate with upregulation of *LIM1* expression by LT in buds, we investigated whether LIM1 plays a role in the regulation of PD dynamics, promoting the opening of PD in response to LT. PD closure is mediated by deposition of callose and conversely, their opening is facilitated by removal of callose and thus high callose is typically associated with closed PDs and low callose levels with open PDs (Bucher et al, 2001; Rinne and Schoot, 1998; Sivaguru et al, 2000). Consequently, change in callose levels serve as useful markers for assessing open or closed PD status. We therefore analyzed callose levels in the buds of LIM1oe and LIM1-RNAi plants both before and after LT treatment (Fig. 3). Both, at 11WSD and after 5 weeks

of LT, callose levels in LIM1oe buds were significantly lower than in WT buds (Fig. 3A,B), while callose levels in LIM1-RNAi buds were significantly higher than in WT buds (Fig. 3C,D). These results suggest that LIM1 negatively regulates callose levels in buds.

In dormant buds, callose levels at PDs are high and consequently cell–cell communication is blocked as PDs are closed (Rinne and Schoot, 1998; Tylewicz et al, 2018). Hence WT hybrid aspen buds fail to reactivate growth under SD even when grafted on FT1-overexpressing root stocks (Tylewicz et al, 2018) since FT1 cannot move due to blockage of cell–cell communication via PD. In contrast, in the hybrid aspen plants expressing *abi1-1* allele with reduced ABA sensitivity, PD are open, and therefore buds can reactivate growth under SD when grafted on FT1-overexpressing root stocks (Tylewicz et al, 2018). Thus, bud break assay complements callose analysis and cumulatively provides insight into PD status. Therefore to confirm the results of callose analysis, we used additional readout of PD status by grafting of buds on FT1 overexpressors followed by bud reactivation under non-inductive conditions as described (Tylewicz et al, 2018). To investigate whether enhancing *LIM1* expression results in open PD in buds, as suggested by low callose levels in LIM1oe buds (Fig. 3A), we grafted LIM1oe and control WT shoot apexes after 11WSD onto FT1-overexpressing root stocks, and investigated their ability to undergo bud break during SD. Whereas bud break was not initiated in dormant WT buds, buds of LIM1oe plants grafted on root stocks of FT1 overexpressors could initiate bud break (Fig. 3E). Thus, results of both callose analysis and bud reactivation assay are highly consistent with cell–cell communication not being blocked in LIM1oe buds suggesting open PDs. Based on these results we propose that induction of *LIM1* in dormant buds by LT downregulates callose levels to promote reactivation of intercellular communication via PD opening to promote dormancy release.

## LIM1 is a positive regulator of FT1 and the GA pathway

To understand the function of *LIM1* in more depth, we investigated its potential downstream targets. Previous work has shown that LT induces expression of *FT1* and *GA20-oxidase*, a key GA biosynthesis gene during dormancy release in hybrid aspen buds (André et al, 2022; Rinne et al, 2011; Singh et al, 2019). Genetic evidence also suggests that *FT1* is required for bud break and GA promotes bud break (André et al, 2022; Singh et al, 2019). Whereas *SVL* has been shown to repress *FT1* and GA pathway in hybrid aspen (Hsiang et al, 2021; Singh et al, 2018), the upstream acting transcription factors that could promote expression of the *FT1* or GA pathway in response to LT are not well characterized. As *LIM1* (like *FT1* and *GA20-oxidase*) is induced in response to LT, we wanted to determine whether *LIM1* mediates LT induction of *FT1* and *GA20-oxidase* expression in the buds. We therefore looked at *FT1* and *GA20-oxidase* expression in LIM1oe and LIM1-RNAi buds before and after LT treatment (Fig. 4). Compared with WT buds, LT activation of *FT1* was higher in LIM1oe. Conversely, in two out of three biological replicates, *FT1* expression in LIM1-RNAi buds was 50% lower than in the wild type, whereas one replicate showed the same expression as wild type after LT. The less pronounced effect of LIM1 downregulation on *FT1* induction by LT (in contrast with significant upregulation in LIM1oe) could be due to LIM1 being downregulated (but not completely suppressed in LIM1-RNAi) with the residual LIM1 may be sufficient to maintain induction of *FT1* expression which makes it difficult to detect strong downregulation in *FT1*. Finally, as *FT1* is

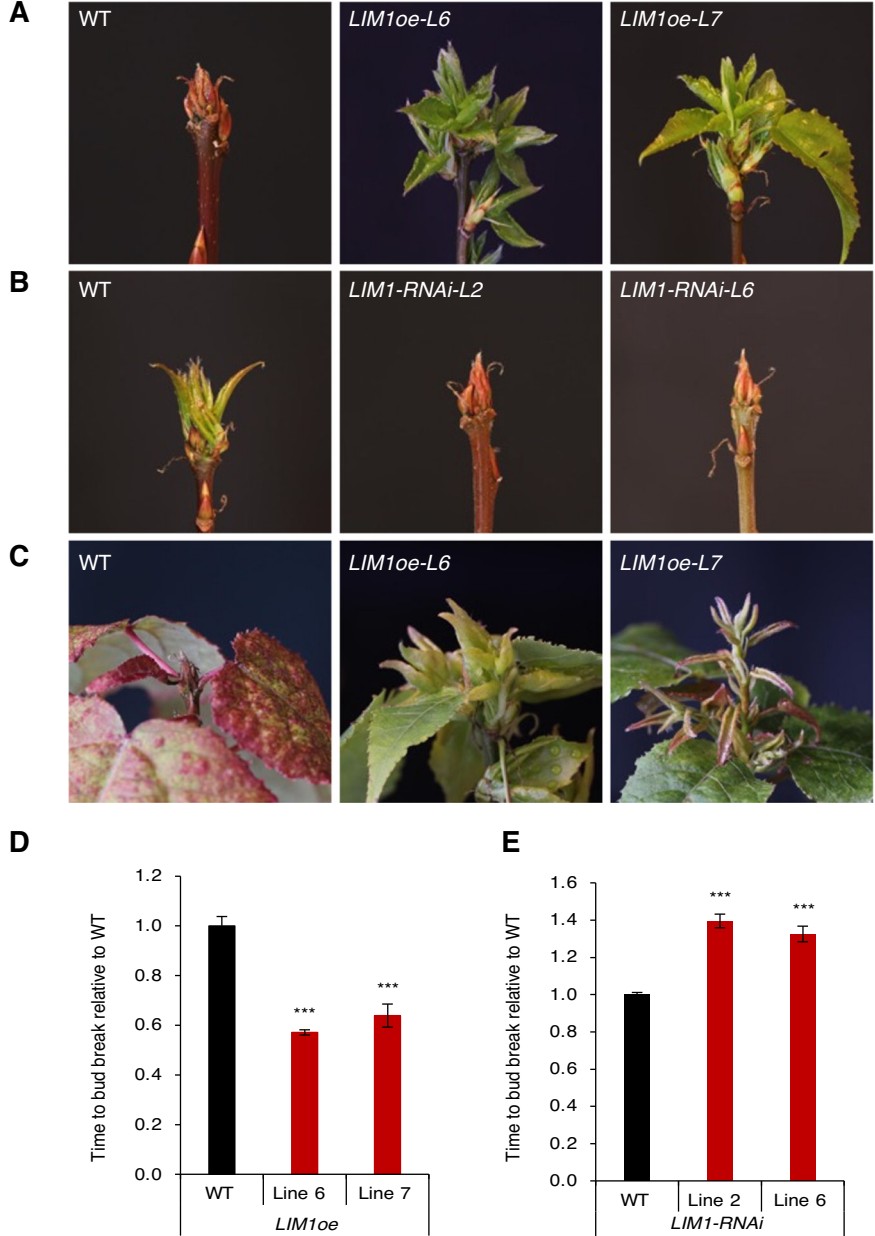

**Figure 2. *LIM1* mediates dormancy release and promotes bud break.**

(A, B) Early and late bud-break phenotypes of *LIM1oe* (A) and *LIM1-RNAi* (B) plants, respectively, compared with wild-type (WT) controls. Plants were first grown under short-day conditions (SD) (16 h dark/8 h light) for 11 weeks, then treated to cold temperatures (4 °C) for 5 weeks, followed by transfer to long-day conditions (LD) (6 h dark/18 h light) for bud burst. (C) Bud break in *LIM1oe* lines moved directly from SD to LD without a cold treatment. (D, E) Quantification of days to bud-break relative to wild-type (WT) controls in *LIM1oe* (D) and *LIM1-RNAi* (E) lines. The average time taken to bud break ± standard error mean (±SEM) is shown, with respect to WT. Statistical analysis was done using unpaired t-test. Asterisks (***) indicate significant difference (***$P < 0.001$) with respect to WT. $P$ values = $3.00 \times 10^{-7}$ (WT vs *LIM1oe-Line 6*), 0.00014 (WT vs *LIM1oe-Line 7*), 0.0014 (WT vs *LIM1-RNAi-Line 2*), and 0.0048 (WT vs *LIM1-RNAi-Line 6*). Source data are available online for this figure.

expressed to very low levels, detection of its downregulation is more challenging (compared to its upregulation). Nevertheless, the overall trend indicates lower FT1 expression in *LIM1* downregulated buds after LT treatment (Fig. 4A). In hybrid aspen, GA20-oxidase genes *GA20-oxidase 1* and *GA20-oxidase 2* are induced in response to LT and we investigated their expression in LIM1oe and LIM1RNAi. Of the two *GA20-oxidase* genes, significantly higher levels of *GA20-oxidase 1*

expression were induced in LIM1oe, and lower levels in LIM1-RNAi buds, after LT (Fig. 4B) whereas the expression of the other *GA20-oxidase* while showing similar pattern was not affected significantly in LIM1oe or LIM1RNAi (Fig. EV1). Based on these results we focused our subsequent analysis on GA20-oxidase 1 (hereafter referred to as GA20-oxidase) and investigated if LIM1 can bind to its promoter. LIM1 could bind the promoter sequence of GA20-oxidase in yeast one-hybrid assay

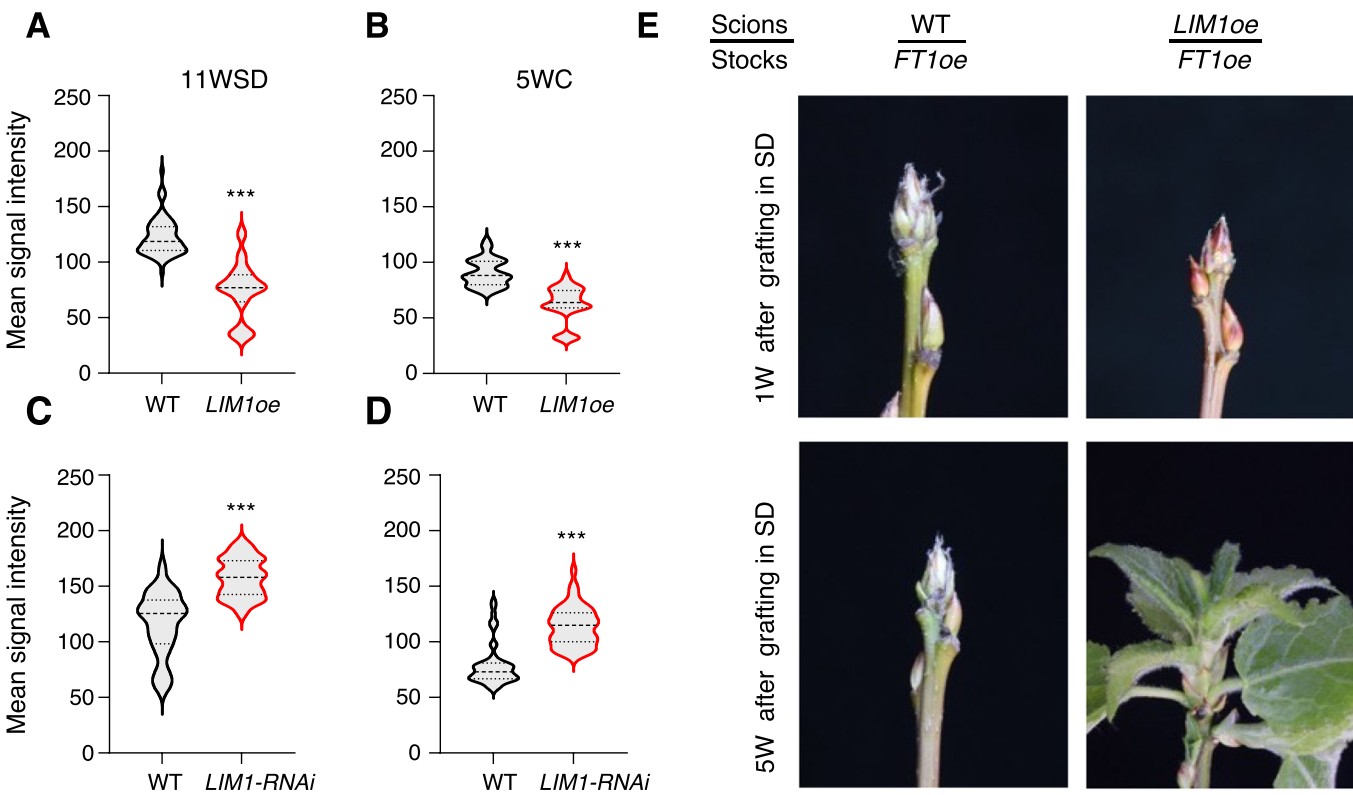

**Figure 3. *LIM1* suppresses callose accumulation to mediate plasmodesmata (PD) opening.**

(A–D) Callose accumulation in *LIM1oe* (A, B) and *LIM1-RNAi* (C, D) buds compared with a wild-type (WT) control after 11 weeks of short-day conditions (SD) (A, C) and 5 weeks of cold treatment (B, D). Callose deposition was examined by measuring aniline blue fluorescence intensity using a confocal microscope. Quantification was carried out using ImageJ, based on 50–60 cells per genotype. Statistical analysis was done using unpaired t-test. Asterisks (***) indicate significant difference (***$P < 0.001$) with respect to WT. $P$ values = $4.36 \times 10^{-22}$ (A), $1.05 \times 10^{-19}$ (B), $2.42 \times 10^{-13}$ (C), and $7.88 \times 10^{-16}$ (D). (E) *FT1*-expressing stocks can reactivate growth in *LIM1* scions under SD. WT and *LIM1* scions (buds) after 11 weeks of SD were grafted onto *FT1*-expressing stocks and kept under SD conditions. Buds remained dormant in WT scions but burst in *LIM1* scions. The pictures of representative graft scions were taken after 1 week (1 W) and 5 weeks (5 W). Source data are available online for this figure.

(Fig. 4C). We then confirmed the interaction of LIM1 with *GA20-oxidase* promoter in buds expressing myc-tagged LIM1 (Myc-LIM1), as revealed by chromatin immunoprecipitation (ChIP)-polymerase chain (Fig. 4D). In contrast with *GA20-oxidase*, we did not observe any binding of Myc-LIM1 in *FT1* promoter at the potential MADS-box binding site (Appendix Fig. S5) This suggests that inducing *LIM1* positively regulates *FT1* and *GA20-oxidase* expression in response to LT. Whereas LIM1 effect on *FT1* expression could be indirect, LIM1 is a direct upstream regulator of GA20-oxidase, and positively regulates the GA pathway by activating the expression of *GA20-oxidase*.

## GA pathway negatively regulates callose accumulation

Exogenous application of GA can induce bud break in axillary buds, which coincides with the removal of dormancy sphincters associated with closed PD (Rinne et al, 2011). However, evidence for role of GA in PD opening in vivo is lacking. As *LIM1* positively regulates the expression of GA20-oxidase, a key enzyme in GA biosynthesis, we examined whether GA is an in vivo modulator of callose accumulation in apical buds. We analyzed the callose levels in the buds of transgenic hybrid aspen overexpressing GA20-oxidase (with elevated GA levels) (Eriksson et al, 2000),

GA2-oxidase-overexpressing plants (with lower GA levels) (Singh et al, 2018), and WT control plants, after 11WSD and 5 weeks of LT. The GA20oxoe plants showed significantly reduced callose levels in SD-induced buds compared with WT (Fig. 5A), and moreover callose levels were also significantly lower in the LT-treated buds of GA20oxoe plants compared with WT (Fig. 5B). Conversely, buds in plants expressing GA2-oxidase had significantly higher callose levels compared with WT after 11WSD (Fig. 5C), as well as after LT (Fig. 5D). Thus GA appears to be a negative regulator of callose accumulation, suggesting that activation of the GA pathway in response to LT can promote PD opening by suppressing callose accumulation which is further supported by results from studies relying on exogenous application of GA to axillary buds (Rinne et al, 2011).

## *LIM1* and *FT1* act along partially redundant pathways in buds

*LIM1, FT1,* and *GA20-oxidase* are all induced by LT, and *LIM1* acts upstream of both *FT1* and *GA20-oxidase*. Functional analyses also identified a role for *LIM1, FT1,* and the GA pathway in the LT response of buds and promoting bud break (this study; (André et al,

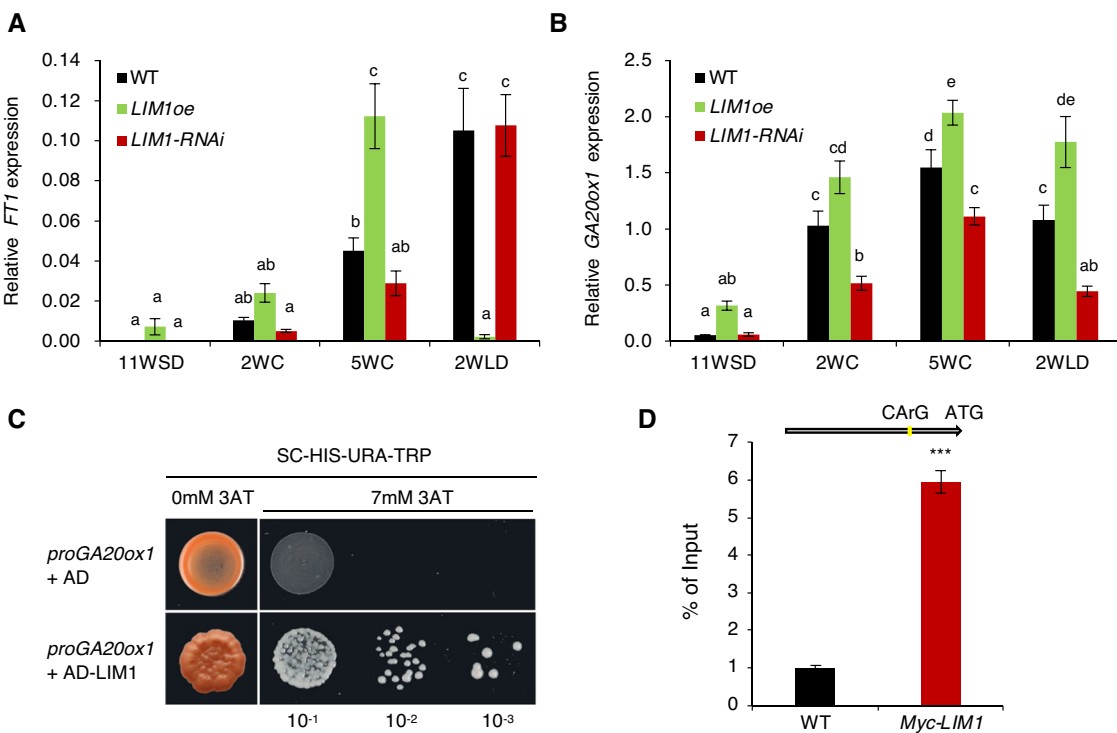

**Figure 4. LIM1 is a positive regulator of FT1 and the gibberellic acid (GA) pathway.**

(A, B) Relative expression of *FT1* (A) and *GA20ox* (B) in buds of wild-type (WT), *LIM1oe* and *LIM1-RNAi* plants at 11 weeks short days (11WSD), 2 (2WC) and 5 weeks of cold (5WC) and after 2 weeks of warm temperature (20 °C) in long days (2WLD). The expression values are relative to the reference gene UBQ and the average of three biological replicates. Error bars indicate standard error mean (±SEM). Different letters over the bars indicate statistically significant differences at *P* < 0.05 by one-way ANOVA Duncan's test. (C, D) LIM1 binds to the *GA20ox* promoter in vitro and in vivo. (C) A yeast one-hybrid assay showing the binding of LIM1 to the *GA20ox* promoter. An empty pDEST22 vector was used as a negative control (AD). (D) Enrichment of the DNA fragment containing the CArG motif in the *GA20ox* promoter quantified by chromatin immunoprecipitation (ChIP)-quantitative polymerase chain reaction (q-PCR). The diagrammatic representation shows the CArG motif present in the *GA20ox* promoter. ChIP-purified DNA was used to perform ChIP-qPCR, expression values are represented as the percentage of input (% of input) DNA. Bars show average values from three independent biological replicates ± SEM. Statistical analysis was done using unpaired t-test. Asterisks (***) indicate significant difference (***P* < 0.001) with respect to WT. *P* value = 0.0002 (WT vs *Myc:LIM1*). Source data are available online for this figure.

2022; Singh et al, 2018). We therefore investigated the genetic interactions between *LIM1*, *FT1*, and the GA pathway to delineate the topology of the LT response network in the buds. Since *LIM1* positively regulates *FT1*, we first inactivated *FT1* in LIM1-overexpressor (LIM1oe/ft1) using crispr-Cas9 approach (Fig. EV2A) and investigated the bud break in LIM1oe/ft1 plants both with and without exposure to LT. Whereas, LIM1oe/ft1 buds were able to undergo bud break as LIM1oe plants after cold treatment (Fig. EV2B), interestingly, loss of *FT1* did not suppress the dormancy phenotype of LIM1oe, as the LIM1oe/ft1 double transgenics displayed the same phenotype as LIM1oe, being able to undergo bud break even without LT treatment after 11 weeks of SDs (Figs. 6A and EV2C). Thus, while *LIM1* positively regulates *FT1* (its downstream target), enhancing *LIM1* expression can bypass the requirement of *FT1* for the LT response in buds.

Because *LIM1* can also positively and directly regulate *GA20-oxidase* (like *FT1*), we took a genetic approach and investigated the effect of reducing GA levels on the bud break phenotype resulting from enhancing *LIM1* expression by generating a transgenic hybrid aspen plant overexpressing *LIM1* and *GA2ox* (LIM1oe/GA2oxoe); with overexpression of GA2-oxidase acting to reduce GA levels (Fig. EV3A). As expected, LIM1oe/GA2oxoe displayed reduced

stature (compared to WT or LIM1oe plants) indicating reduced GA levels in these plants. We then investigated the effect of reducing GA levels on the bud break phenotype as a result of *LIM1* enhancement in LIM1oe/GA2oxoe compared with single LIM1oe and WT plants. While LIM1oe displayed early bud break, expression of GA2-oxidase resulted in suppression of the early bud break phenotype in LIM1oe/GA2oxoe, reverting almost to WT (Fig. EV3B). This result further confirms that the GA pathway is a downstream target of *LIM1* in buds. Surprisingly, however, a reduction of GA was not sufficient to suppress the dormancy defects resulting from LIM1oe, because unlike WT control buds that did not undergo bud break and remained dormant, LIM1oe/GA2oxoe buds were able to restart growth even when these plants were transferred directly from SD to LD without dormancy-releasing LT as observed for LIM1oe plants (Figs. 6B and EV3C).

LIM1 positively regulates the expression of *FT1* as well as of *GA20-oxidase*, the key enzyme in GA biosynthetic pathway. However, inactivating *FT1* or reducing GA levels individually did not suppress the dormancy defects of LIM1oe. Therefore, we explored the possibility of redundancy in LT pathways involving *LIM1*, *FT1*, and GA in bud dormancy regulation. To test this, we blocked GA biosynthesis in the LIM1oe/ft1 background using

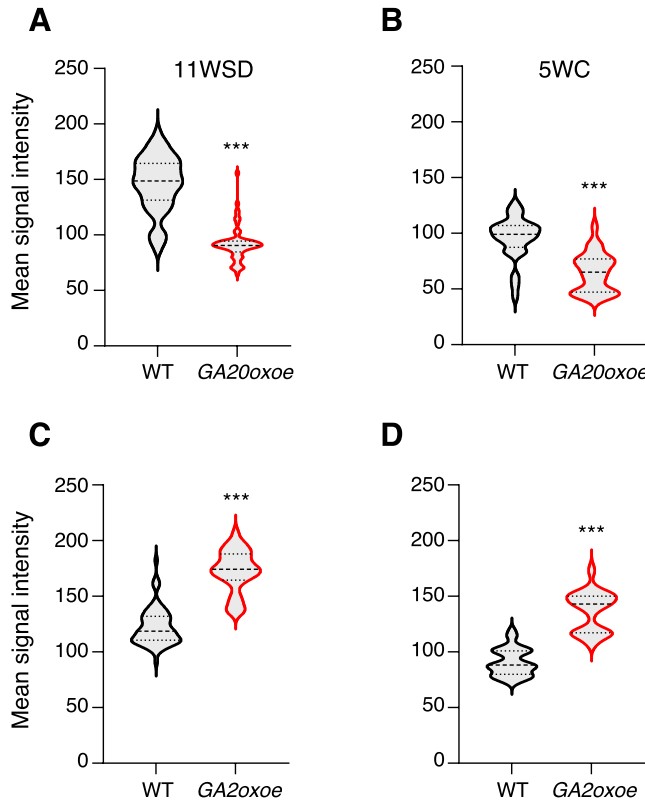

**Figure 5. The gibberellic acid (GA) pathway negatively regulates callose accumulation.**

(A–D) Callose accumulation in *GA20oxoe* (A, B) and *GA2oxoe* (C, D) buds compared with a wild-type (WT) control after 11 weeks of short-day conditions (SD) (A, C) and 5 weeks of cold treatment (B, D). Callose deposition was examined by measuring aniline blue fluorescence intensity using a confocal microscope. Quantification was carried out using ImageJ, based on 50–60 cells per genotype. Statistical analysis was done using unpaired t-test. Asterisks (***) indicate significant difference (***$P < 0.001$) with respect to WT. P values = $1.13 \times 10^{-26}$ (A), $1.78 \times 10^{-13}$ (B), $1.62 \times 10^{-30}$ (C), and $1.89 \times 10^{-31}$ (D). Source data are available online for this figure.

paclobutrazol (a GA biosynthesis inhibitor) to simultaneously block FT1 and the GA pathway and then investigated dormancy and bud break. In contrast to LIM1oe/ft1 and LIM1oe/GA2-oxidase plants that individually displayed same dormancy defects as LIM1oe (i.e., bypassing the requirement of cold for dormancy release) (Fig. 6A,B), simultaneously inactivating FT1 and blocking GA biosynthesis completely suppressed the dormancy defects in LIM1oe/ft1 plants (Fig. 6C). Upon paclobutrazol treatment, LIM1oe/ft1 plants did not undergo bud break when transferred directly from SD to LD without cold treatment (Figs. 6C and EV4A), nor did paclobutrazol treated LIM1oe/ft1 buds undergo bud break after cold treatment (Fig. EV4B,C). The complete blockage of bud break in paclobutrazol treated LIM1oe/ft1 (in contrast to delay in bud break in LIM1oe/GA2oxoe) even after cold treatment indicates that blocking both FT1 and GA pathway simultaneously, prevents the release of dormancy itself, which then is reflected in a failure to undergo bud break in these plants. This analysis suggests that enhancing *LIM1* can bypass a lack of *FT1* as it can activate GA pathway directly. Together with the gene expression data and

genetic analysis, this suggests that GA (via regulation of GA20-oxidase expression) is the shared downstream target of *LIM1* and *FT1* in buds with *LIM1* and *FT1* acting in a partially redundant pathway, converging on GA20-oxidase in the LT response of buds. This suggestion is also consistent with the rescue of the bud break phenotype in *ft1* mutant by exogenous application of GA (Sheng et al, 2023), which also suggests that *FT1* (like *LIM1*) is upstream of GA pathway.

## Negative feedback modulates the *LIM1* response in buds to LT

Our data identified *LIM1* as a crucial regulator of the bud response to LT. As LIM1 is a potent promoter of dormancy release and bud break, *LIM1* expression needs to be tightly regulated so that it increases gradually rather than rapidly, scaling with exposure to LT, and linking LT exposure with the gradual release of dormancy. To understand the mechanisms behind the gradual increase of *LIM1* expression in response to LT, we created a mathematical model to simulate the network of key interactions (Fig. 7A, solid lines). The model comprised a system of ordinary differential equations (ODEs) that represented the dynamics of *LIM1*, *FT1*, *SVL*, *GA20-oxidase,* and *GA* (see Appendix Methods). Based on gene expression presented here (Fig. 4) as well from Singh et al (2018), we simulated LIM1 promoting *FT1* expression, SVL inhibiting *FT1* expression (Singh et al, 2018), and both LIM1 and FT1 promoting *GA20-oxidase* expression (this study, and André et al, 2022), with an increase in GA20-oxidase levels promoting GA synthesis (Eriksson et al, 2000). After transfer to LT, the expression of *LIM1* increased and *SVL* decreased over a period of weeks (Fig. 1) (Singh et al, 2018). We prescribed Hill functions to represent these gradual changes in *LIM1* and *SVL*, selecting parameters that enabled the model predictions to agree with the data (Appendix Table S1 and see Appendix Methods). Signaling networks typically equilibrate over a time scale of hours (Alon, 2006). However, since our studies were on a much longer time scale, we assumed that the interaction network would be in a steady state during the weeks-long observation of *LIM1* and *SVL*. Overall, the model simulations demonstrated that the proposed interaction network was sufficient to explain the observed dynamics after transfer to LT. With estimated parameter values (listed in the Appendix Table S1), the predictions were in good agreement with the measured changes in dynamic expression in WT, LIM1oe, and LIM1-RNAi plants (blue lines in Fig. 7B, Appendix Fig. S6A). One exception to this was FT1 upregulation in LIM1oe which could hint at additional mechanisms such as epigenetic regulation that have been shown to mediate in LT response of gene expression in buds (Sato and Yamane, 2024) could also contribute to FT1 regulation and were not included in the modeling.

Thus, while the mathematical model could reproduce the experimental data, a critical requirement for the model to function was that *LIM1* upregulation is gradual which is also observed experimentally (Fig. 1). This led us to investigate the possible mechanisms that could facilitate a gradual upregulation of *LIM1* that is crucial for dormancy regulation. Negative feedback is a regulatory motif known to dampen rapid upregulation and provide robust regulation to signaling pathways (Alon, 2007). Therefore, we tested this possibility by introducing negative feedback into the model. Typically, the final product or output of a pathway provides this negative feedback, and therefore, in the case of *LIM1*, we modeled negative feedback from GA, the final output of the *LIM1* pathway

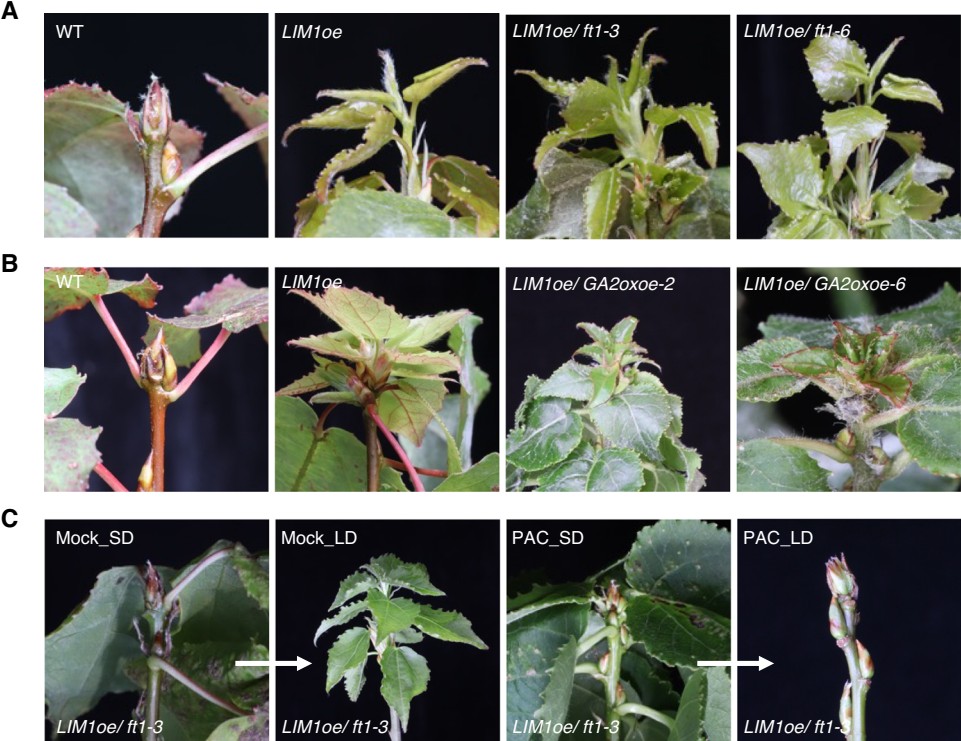

**Figure 6. *LIM1* and *FT1* function in a partially redundant manner that converges on the gibberellic acid (GA) pathway.**

(A, B) Bud-break phenotypes of WT, *LIM1oe*, and *LIM1oe/ft1* (A) and WT, *LIM1oe*, and *LIM1oe/GA2oxoe* (B) plants moved directly from 11 weeks of short-day conditions (SD) to long-day conditions (LD) without cold treatment. (C) Bud-break phenotypes of *LIM1oe/ft1* plants treated with Mock and paclobutrazol (PAC) for 11 weeks of SD then moved directly to LD without cold treatment but continued with Mock and PAC treatments. Source data are available online for this figure.

(the red dashed inhibitory arrow in Fig. 7A) and tested if this could contribute to gradual upregulation of *LIM1*. Interestingly, introducing a negative feedback from GA to *LIM1* indeed enabled a gradual increase in *LIM1* (orange line in Fig. 7B), even assuming relatively rapid *LIM1* upregulation by LT. Conversely, the absence of negative feedback would lead to a rapid increase in *LIM1* expression already at the onset of LT (green line in Fig. 7B). Thus, with negative feedback the model could match the experimental data even under the scenario when LT induction of LIM1 was assumed to be rapid (Fig. 7B; Appendix Fig. S6B) and consequently, this also resulted in a more gradual increase in expression of *GA20-oxidase*.

We then tested experimentally whether GA negatively feeds back on *LIM1* expression, by investigating *LIM1* expression in buds with enhanced GA levels. The data indicated significantly lower expression of *LIM1* in buds of GA20oxe, suggesting a strong negative feedback by GA on *LIM1* expression (Fig. 7C). Thus, while other mechanisms could contribute to a gradual *LIM1* upregulation, as suggested by modeling, experimental validation of the predicted negative feedback suggests that negative feedback can contribute to the observed gradual LIM1 increase in vivo. LIM1 is a crucial upstream regulator of *FT1* and GA, the two key components of LT response. Consequently, having a negative feedback would facilitate that dormancy release is appropriately timed by guarding against rapid *LIM1* upregulation which otherwise would lead to altered dormancy and bud break regulation. Thus modeling and experimental data suggests that negative feedback on LIM1 could plausibly contribute to robust regulation of dormancy release and bud break.

## Discussion

Environmental and hormonal regulation of PDs is a conserved mechanism implicated in diverse biological processes e.g., dormancy (Rinne and Schoot, 1998; Tylewicz et al, 2018; Zhao et al, 2023), tuberization (Nicolas et al, 2022), slow to fast growth switch in lily buds (Pan et al, 2023) and root branching in response to change in water availability (Mehra et al, 2022). In tree buds, opening of PDs is crucially associated with dormancy release and is induced by exposure to LT. Transcription factors of MADS-box family have been implicated in dormancy regulation in several tree species and those of *SVL/DAM* family have been implicated in PD closure (Singh et al, 2019; Zhao et al, 2023). In contrast, mediators of PD opening in response to LT are not well studied. We have identified LIM1 transcription factor as an upstream regulator of FT1 and GA pathway that mediates the opening of PD in buds in response to LT. LIM1 thus links the temperature and hormonal control of cell–cell communication with dormancy release and subsequent bud break.

### *LIM1*- LT-induced mediator of dormancy release

PD closure plays a key role in dormancy induction, while LT promotes opening of PD and dormancy release. The transcriptional and hormonal networks that promote PD closure have been characterized (Singh et al, 2019; Tylewicz et al, 2018; Zhao et al, 2023), but the regulators of LT-induced PD opening are not as well understood. Gene

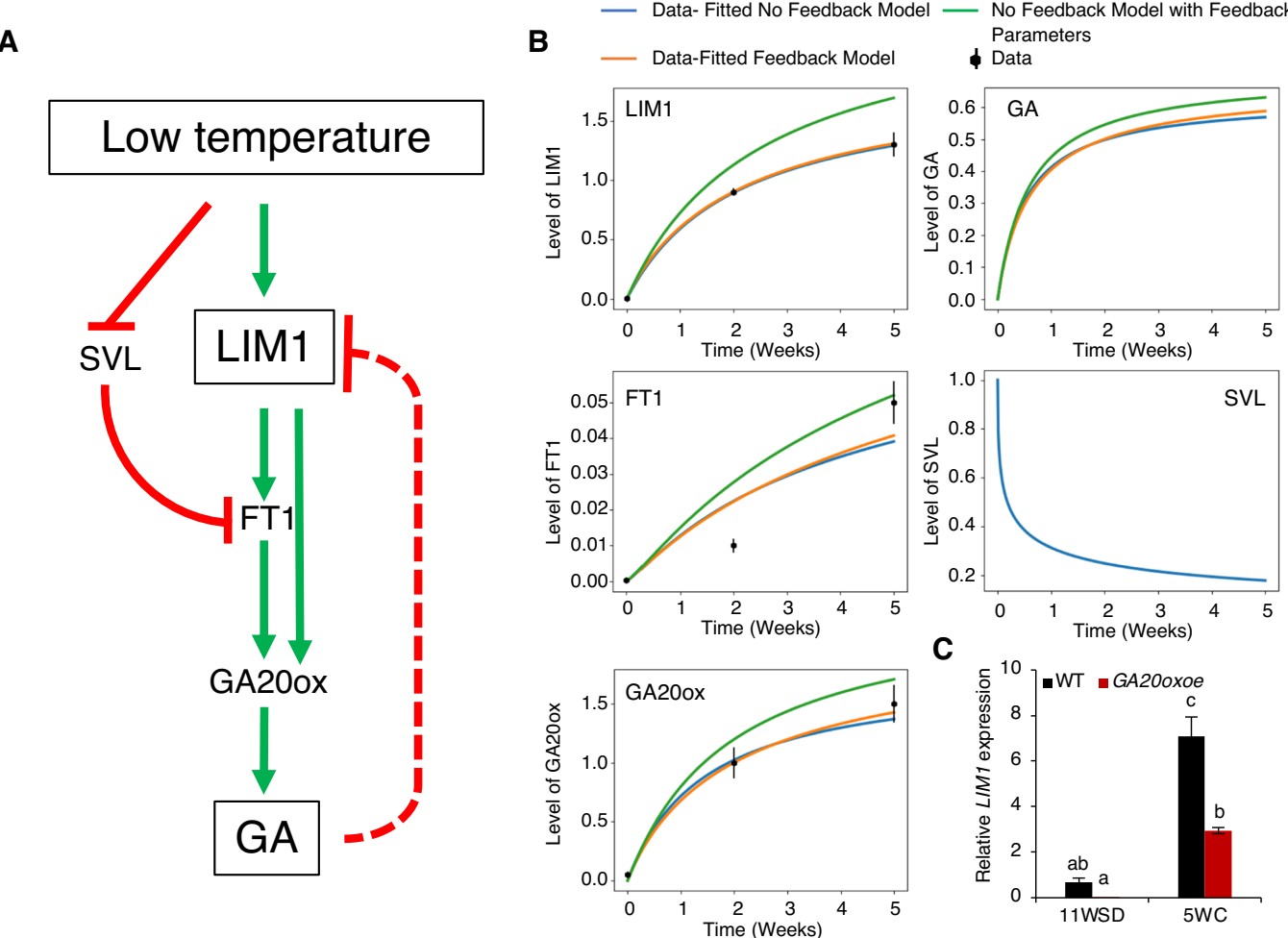

**Figure 7. GA-mediated feedback regulation ensures a gradual increase in *LIM1* expression.**

(A) A proposed genetic network for LT response of buds in dormancy and subsequent bud break. Green arrows indicate positive regulation and red block lines indicate negative regulation. The abbreviations used are SHORT VEGETATIVE PHASE-LIKE (SVL), LOW TEMPERATURE INDUCED MADS-BOX 1 (LIM1), FLOWERING LOCUS T (FT1) GA20-oxidase (GA20ox) and GA (Gibberellin). (B) Model predictions for the dynamics of LIM1, FT1, GA20ox, GA, and SVL after the onset of LT, using estimated parameters given in Appendix Table S1. Lines show the predictions with no feedback (blue), feedback whereby GA inhibits LIM1 (orange) and no feedback albeit using the parameters estimated from the model with feedback (green). Dots and error bars show the LIM1, FT1, and GA20ox experimental data for wild-type (as in Figs. 1 and 4). (C) Negative feedback of GA on LIM1 expression in buds. LIM1 expression in buds of wild type and GA20-oxidase plants before (10WSD) and after cold treatment 4 WC) subjected to cold. The expression values are relative to the reference gene UBQ and the average of three biological replicates. Error bars indicate standard error mean (±SEM). Different letters over the bars indicate significant differences at $P < 0.05$. Statistical analysis was done using one-way ANOVA implying Dunnett's/Tukey's multiple comparison test. Source data are available online for this figure.

expression studies in hybrid aspen and other plants have identified several transcription factors, as well as genes such as *FT1* and those of the GA biosynthesis pathway, e.g., *GA20-oxidase*, that are upregulated in buds in response to LT (André et al, 2022; Rinne et al, 2011). FT1 and GA appear to play a positive role in bud break and possibly dormancy release, although their mode of action is also not well understood. The repressors of *FT1* expression and the GA pathway, e.g., *SVL/DAM* are known (da Silveira Falavigna et al, 2021; Zhao et al, 2023). Intriguingly though, suppression of SVL by LT is not sufficient to fully explain LT response of buds (Brunner, 2021). Thus components other than *SVL/DAM* are likely involved in LT response of buds, for example factors with antagonistic action to *SVL/DAM* presumably acting as activators of the *FT1* and GA pathway. However, such activators mediating LT induction of *FT1* and GA are not as well

studied. We have now identified *LIM1*, a previously uncharacterized MADS-box transcription factor induced by LT in buds and provided evidence for its role in LT response in buds acting antagonistically to repressors such as *SVL/DAM*.

Gene expression and genetic data show that LIM1 functions as a positive upstream regulator of FT1 and GA20-oxidase, providing a genetic link between LT and induction of *FT1* and the GA pathway during dormancy release. The induction of *FT1* and the GA pathway is therefore not only a result of derepression arising from the LT-induced suppression of *SVL/DAM* expression (and related factors), but also caused by LT-mediated induction of activators such as *LIM1*. Importantly, unlike *FT1* or *GA20-oxidase*, *LIM1* expression is not repressed by *SVL* and thus induction of *LIM1* by LT is not due to repression of *SVL* (or induction of *FT1*). These

data indicate upstream pathways controlling *SVL* repression and *LIM1* activation by LT could be distinct, highlighting the complexity of LT response of buds. The induction of *LIM1* by LT, opposing patterns of *LIM1* and *SVL* expression, and dormancy defects caused by enhancing *LIM1* expression, are all consistent with *LIM1* as an LT-induced promoter of dormancy release, acting antagonistically to *SVL*. Based on these results we propose that, when dormant buds sense LT, the induction of *LIM1* with simultaneous repression of antagonistically acting *SVL*, promotes the release of dormancy by activating downstream components such as *FT1* and the GA pathway.

## LT-induced PD opening in buds is mediated by an LIM1-GA module

The LT-induced opening of PD is a pivotal event associated with the release of buds from dormancy. PD are blocked by electron-dense dormancy sphincters and, upon exposure to LT, these PD-associated dormancy sphincters are removed, mirroring dormancy release (André et al, 2022; Rinne et al, 2011; Rinne and Schoot, 1998). Intriguingly, in *ft1* mutant plants, which fail to undergo bud break even after LT treatment (suggesting a possible role of *FT1* in dormancy), the dormancy sphincters are removed in response to LT, as in WT buds (André et al, 2022). This suggests that PD opening can be achieved independently of FT1. Thus, in contrast to the well-characterized regulation of PD closure during dormancy induction, regulation of LT-induced PD opening and potential regulators involved, remain largely unknown. Our data showing negative regulation of callose levels by LIM1 as well as induction of bud break by under non-inductive conditions upon grafting on FT1oe in LIM1oe but not in wild type suggest that enhancing *LIM1* reactivates cell–cell communication. Together with LT induction of *LIM1* correlating with PD opening, our data strongly suggest that LIM1 is a previously unrecognized crucial mediator of PD dynamics and facilitates opening in response to LT in the buds by downregulating callose levels.

LT induces the expression of *GA20-oxidase*, a key enzyme in the GA biosynthesis pathway. Also, exogenous application of GA to axillary buds has also been shown to result in removal of dormancy sphincters (Rinne et al, 2011). However, the in vivo role of GA in PD opening in apical buds has not been demonstrated. Our data shows that enhancing GA20-oxidase expression led to low callose levels, and conversely GA2-oxidase overexpression led to high callose. These results demonstrate that GA is a negative regulator of callose levels, establishing its role as an endogenous promoter of PD opening in response to LT in apical buds. Importantly, upstream activators of the GA pathway that could link GA with PD opening and its regulation by LT are not well studied. Thus, showing *LIM1* as the LT-induced upstream regulator of the GA20-oxidase expression, provides a crucial genetic link between LT, activation of the GA pathway and PD opening in the buds.

## LIM1 and FT1 pathways converge on shared targets in LT response of buds

Expression analyses and functional studies have so far identified *FT1* and the GA pathway as two major candidates for mediating the LT response of buds (André et al, 2022; Rinne et al, 2011; Sheng et al, 2023). Intriguingly, despite *FT1* being the downstream target of *LIM1*,

inactivating FT1 in LIM1oe was not sufficient to suppress bud break in LIM1oe plants. Similarly, reducing GA levels also did not suppress dormancy defects resulting from enhanced *LIM1* expression. These observations lead us to propose that *LIM1* and *FT1* act along partially redundant pathways in LT response of buds as summarized in the network (Fig. 7A). Experimental data showing inactivating *FT1* or reducing GA separately, does not suppress LIM1oe phenotype whereas simultaneous inactivation of FT1 and reducing GA levels does suppress LIM1oe dormancy phenotype is consistent with the topology of the proposed network (Fig. 7). Also, since *FT1* and *LIM1* converge on GA pathway, enhancing *LIM1* expression can presumably bypass the lack of *FT1* in LIM1oe/ft1 by directly activating the GA pathway, their shared downstream target, which can both activate PD opening and promote bud break. In contrast, in the single *ft1* mutant, endogenous activation of *LIM1* by LT is presumably not sufficient (unlike overexpression of *LIM1* in the double transgenic LIM1oe/ft1) to activate the GA pathway downstream to either release dormancy and/or promote bud break. In accordance with this, exogenous application of GA to an *ft1* mutant can suppress the mutant phenotype (Sheng et al, 2023), as predicted by the proposed network. These results thus reveal the topology of the network mediating the LT response of buds via *FT1* and *LIM1*, forming what appears to be a coherent feedforward loop (Alon, 2007), converging on GA in the LT response.

## LIM1 has a dual role in dormancy and bud break

From our data, it appears that *LIM1* could either be specifically involved in dormancy release (via opening of PD) and only impact bud break indirectly through its effect on dormancy release, or have a role in bud break in addition to its role in dormancy release. The reduction in *LIM1* expression during the switch to bud break would argue for an exclusive role of *LIM1* in dormancy release, with only an indirect effect on bud break as a consequence of this function. However, our results argue against this. For example, *LIM1* expression, despite downregulation, is much higher in buds poised for bud break than in dormant buds. Moreover, our data shows that *LIM1* positively regulates expression of *FT1* and GA20-oxidase, the key component of GA pathway which have a promotive role in bud break (Gao et al, 2024; Singh et al, 2018) as well. These results lead us to favor dual roles for *LIM1*, in LT-mediated dormancy release by LT and subsequently in promoting bud.

Intriguingly, in contrast to positive effect of LIM1oe on *FT1* expression in LT-treated buds, *FT1* expression was significantly reduced in LIM1oe after transfer from LT to warm temperature. It has been shown that *FT1* expression is rapidly downregulated in late stages of bud break (André et al, 2022) and since bud break is significantly advanced in LIM1oe (compared to wild type, Fig. 4A), *FT1* downregulation may reflect advanced bud break in LIM1oe compared to wild type. In addition, it is also possible that there is a switch in LIM1 activity from activator to repressor during bud break which may contribute to repression of *FT1* in LIM1oe. The MADS-box transcription factors are known to heterodimerize which can modulate the properties of resulting complexes (Puranik et al, 2014). Interestingly, in apple buds, diverse MADS-box complexes are associated with distinct stages of dormancy and bud break (da Silveira Falavigna et al, 2021). Thus, it is possible that LIM1 may interact with MADS-box proteins during bud break that are distinct from those during dormancy release which could contribute to such a switch in LIM1 activity during bud break.

## Negative feedback by GA in the LT regulation of *LIM1*

It is worth noting that *LIM1* expression scales with exposure to LT, i.e., the longer the exposure of buds to LT, the higher the expression of *LIM1*. Given the function of *LIM1* as a crucial regulator of PD opening and bud break, this scaling of *LIM1* expression could provide a link between exposure of buds to LT, PD opening and temporal control of dormancy release. As a potent promoter of dormancy release and promoting bud break, it is therefore essential that *LIM1* expression is upregulated gradually in response to LT, as too rapid an increase of *LIM1* expression by LT could lead to premature dormancy release and precocious bud break, compromising survival of buds.

Mathematical modeling (Fig. 7B) revealed that negative feedback could contribute to a dampening of the rapid response of *LIM1* to LT and subsequent gene expression analysis corroborated a negative feedback between GA and *LIM1* expression in buds (Fig. 7C). Cumulatively, the modeling and experimental evidence indicates that negative feedback is a plausible mechanism, mediating the appropriate regulation of *LIM1* expression dynamics by LT to ensure a more gradual increase in *LIM1* expression. This would facilitate precise alignment of the temperature input for a robust temporal regulation of dormancy release. In addition, negative feedback loops are associated with homeostasis and reduce the impact of perturbations of noisy input signals (Alon, 2007). The release of dormancy and bud break are both regulated by fluctuating temperature cues, and the negative feedback of GA on *LIM1* regulation could play a crucial role in buffering the impact of temperature variations on dormancy regulation. While we find evidence for negative feedback regulation of *LIM1*, additional factors e.g., epigenetic regulation may also contribute to precise *LIM1* response to LT as has been observed in case of LT regulation of MADS-box factors such as *FLC-like* gene in axillary buds of Kiwi or *DAM* genes in buds of various tree species (Voogd et al, 2022).

## Hormonal control of cell–cell communication—a non-canonical regulatory mechanism

Dormancy establishment and dormancy release (and subsequent bud break), are governed by two major environmental cues (photoperiod and temperature). Antagonistically acting ABA and GA are the key hormonal mediators of these two cues in dormancy regulation. Interestingly, antagonism between ABA and GA has been exploited also in several processes including seed dormancy regulation and involves crosstalk at the level of biosynthesis, gene expression and transport (Liu and Hou, 2018; Topham et al, 2017; Vanstraelen and Benková, 2012; Weiss and Ori, 2007). Our results show yet another level at which ABA/GA antagonism acts, converging on the regulation of cell–cell communication.

Hormonal control of PDs to regulate cell–cell communication appears to widely used mechanism that extends beyond tree dormancy regulation. For example, ABA and more recently brassinosteroids have been shown to mediate in closure of PDs in Arabidopsis roots (Mehra et al, 2023; Wang et al, 2023). Contrastingly, our results, show in vivo role for GA in opening of PDs. Our results showing that GA levels can modulate PD dynamics in vivo by controlling callose levels, has wider implications for the GA-mediated control of development via regulation of cell–cell communication. For example, labeled GA has been shown to move via PD in buds (Rinne and Schoot, 1998) and

thus GA can presumably regulate its own distribution via dynamic regulation of PDs. In addition, by regulating PD dynamics, GA could potentially influence the distribution/transport of other regulatory molecules that move via PD. Thus GA-mediated control of cell–cell communication could be a further, hitherto unrecognized, non-canonical GA-mediated regulatory mechanism in plant development.

Temperature is a major environmental cue regulating developmental transition in plants. In contrast with rapid responses to change in temperature, major developmental transitions such as vernalization or bud dormancy involve long-term sensing of temperature input (Antoniou-Kourounioti et al, 2021). Long-term sensing of low temperature during vernalization is mediated by epigenetic silencing of *FLC*, a floral repressor in Arabidopsis (Bastow et al, 2004; Sung and Amasino, 2004). Whereas long-term exposure to LT regulates bud dormancy in which control of cell–cell communication, plays an important role (Rinne et al, 2011; Rinne and Schoot, 1998). Our results further reveal the complexity of this LT response in buds and show that *LIM1* and *FT1* form a coherent feedforward loop in dormancy release and negative feedback via GA on *LIM1*, that can contribute to robust regulation of bud dormancy by LT. Thus, by identifying the role of the LIM1-GA module in mediating PD opening, we have provided an important insight into the molecular regulation of dormancy and the mechanism underlying seasonally aligned growth in trees by temperature.

# Methods

**Reagents and tools table**

| Reagent/Resource | Reference or Source | Identifier or Catalog Number |
|---|---|---|
| **Experimental Models** | | |
| Hybrid aspen (*Populus tremula x Populus tremuloides*) | Umea Plant Science Centre | N/A |
| DH5α (*E. coli*) | Thermo Fischer Scientific | EC0112 |
| GV3101 (Agrobacterium) | Gold Biotechnology | CC-207 |
| YM4271 and Yα1867α (Yeast strains) | Reece-Hoyes and Walhout, 2018 | N/A |
| **Recombinant DNA** | | |
| pK2GW7 | VIB-UGENT | Vector ID: 1_08 |
| LIM1-pK2GW7 (LIM1oe) | This study | N/A |
| pK7GWIWG2 (I) | VIB-UGENT | Vector ID: 1_28 |
| LIM1-pK7GWIWG2(I) (LIM1-RNAi) | This study | N/A |
| pGWB18 | Nakagawa's lab | N/A |
| pGWB18_Myc-LIM1 | This study | N/A |
| pHSE401 | Addgene | 62201 |
| pCBC-DT1T2 | Addgene | 50590 |
| FT1-pHSE401 | This study | N/A |
| pDONR P4-P1R | Thermo Fischer Scientific | N/A |
| PMW2 | Addgene | 13349 |
| pDEST22 | Thermo Fischer Scientific | N/A |

| Reagent/Resource | Reference or Source | Identifier or Catalog Number |
|---|---|---|
| **Antibodies** | | |
| Anti-Myc monoclonal antibody | Abcam | ab32; GR255064 |
| **Oligonucleotides and other sequence-based reagents** | | |
| PCR Primers | This study | Appendix Table S2 |
| **Chemicals, Enzymes and other reagents** | | |
| Phusion High-Fidelity DNA Polymerase | Thermo Fischer Scientific | F530S |
| T4 DNA Ligase | NEB | M0202S |
| ECO31l | Thermo Fischer Scientific | FD0294 |
| Paclobutrazol | MCE | HY-B0853 |
| **Software** | | |
| MEGA | https://www.megasoftware.net/mega4/index.php | N/A |
| ImageJ | https://imagej.net/software/imagej/ | N/A |
| Graphpad Prism | https://www.graphpad.com/features | N/A |

## Plant material and growth conditions

Wild-type (WT) hybrid aspen (*Populus tremula × tremuloides*, clone T89) and transgenic plants were grown on half-strength Murashige–Skoog medium (Duchefa) under sterile conditions for 4 weeks, then transferred to soil and cultivated with fertilization for 5 weeks in a greenhouse (18 h, 22 °C day/6 h, 18 °C night cycle). Plants were then transferred to short-day conditions (SD) (8 h, 20 °C light/16 h, 15 °C dark cycles) for 11 weeks, to establish growth cessation and dormancy. The response to SD was determined by monitoring bud set and plant growth. After 11 weeks of SD in the growth chambers, the plants were exposed to low temperatures (4 °C) for 5 weeks, to release dormancy, and then LD and warm-temperature conditions (LD/WT) for bud break. Bud break was characterized by bud swelling and the emergence of green leaves. Apices of WT and transgenic plants were sampled for gene expression analysis after the plants had ceased growth and developed dormancy, i.e., after 11 weeks of SD (11WSD), after exposure to the cold (4 °C) for 2 weeks (2WC) and 5 weeks (5WC) to induce dormancy release, and two weeks after the transfer to LD/WT conditions (2WLD). Each sample was immediately frozen in liquid nitrogen and stored at −80 °C until further use. Pictures of apices were taken using a Canon EOS digital camera to monitor bud break.

## Generation of plasmid constructs

To generate LIM1oe and LIM1-RNAi constructs, a full-length *Low temperature Induced MADS-box 1* (*LIM1*) coding sequence (CDS) and 291-bp CDS fragments, respectively, were amplified by polymerase chain reaction (PCR) from cDNA prepared using

mRNA extracted from hybrid aspen plants as templates, using the primers listed in Appendix Table S2. Both full-length *LIM1* and *RNAi* DNA fragments were cloned into pENTR/D-TOPO vectors and then sequenced, followed by sub-cloning into the plant transformation vectors pK2GW7 and pK7GWIWG2(I) (Karimi et al, 2002) containing a CaMV35S promoter to generate LIM1-pK2GW7 (LIM1oe) and LIM1-pK7GWIWG2(I) (LIM1-RNAi) constructs.

The Myc-LIM1 fusion construct was generated by cloning a full-length LIM1 coding sequence into the pGWB18 plant transformation vector containing a CaMV35S promoter and 4xMyc tag at the N-terminal. To generate FT1crispr (*ft1*) mutants, guide RNAs 1 (TGCGAGCTCAAACCCTCTCAGG) and 2 (GCCGAGGGTTGA-TATTTGGCGGG) were designed using the online CRISPR-P tool (http://crispr.hzau.edu.cn/CRISPR2/), and cloning was performed following (Xing et al, 2014). The primers listed in Appendix Table S2 were used to amplify the template from pCBC-DT1T2, and the PCR product was cloned into the pHSE401 binary vector using the Golden Gate approach.

## Plant transformation and screening of transgenic lines

WT hybrid aspen plants were transformed with LIM1-pK2GW7, LIM1-pK7GWIWG2(I), and FT1-pHSE401 constructs via *Agrobacterium*-mediated transformation (Tylewicz et al, 2015) to generate LIM1oe, LIM1-RNAi and *ft1* transgenic lines. To generate GA2oxoe/LIM1oe and LIM1oe/ft1 lines, LIM1-pK2GW7 and FT1-pHSE401 constructs were transformed into GA2oxoe (Singh et al, 2019) and LIM1oe backgrounds, respectively. Details for the generation of other transgenic lines, such as SVL-RNAi, FT1oe and GA2oxoe, have been described previously (Miskolczi et al, 2019; Singh et al, 2018). For the screening of transgenic lines, total RNA was isolated from a small samples of plants with apices and a few leaves, followed by cDNA preparation and quantitative (q)PCR analysis. FT1crispr (*ft1*) lines were screened by PCR using genomic DNA as a template, with the primers listed in Appendix Table S2. Deletion mutations of FT1 in FT1cripsr (*ft1*) and LIM1oe/ft1 were confirmed by sequencing the PCR products.

## RNA isolation and qRT-PCR analysis

Total RNA was extracted from plant tissues (shoot apices) using the Spectrum™ Plant Total RNA Kit (Sigma-Aldrich). RNA (10 µg) was treated with RNase-free DNaseI (Life Technologies, Ambion), and 1 µg was then utilized for cDNA synthesis using an iScript cDNA Synthesis Kit (BioRad). Ubiquitin was used as the reference gene in all experiments. Quantitative reverse transcription (qRT)-PCR experiments were conducted using a LightCycler 480 SYBR Green I Master mix and a LightCycler 480 II instrument (both supplied by Roche). The Δ-cq method was used to calculate the relative expression values for genes of interest (Vandesompele et al, 2002). Primer sequences used in the qPCR experiments are given in Appendix Table S2.

## Grafting experiment

WT and LIM1oe plants were grown in chambers under short photoperiod conditions (SD) (8 h, 20 °C light/16 h, 15 °C dark cycles, 80% relative humidity) for 11 weeks. Scions of the WT and

LIM1oe plants were then grafted onto root stocks of FT1-overexpressing plants that had developed about ten leaves, as described elsewhere (Nieminen et al, 2008). Grafted plants were kept under SD and monitored for bud break.

## Callose deposition

For callose staining, fresh buds from 11WSD and 5WC plants were longitudinally sectioned (70 μm thickness) using vibratome, and the sections incubated in 0.1% aniline blue (Fluka, www.sigmaaldrich.com/brands/Fluka_Riedel_home.html) solution for 2 h in the dark. After washing twice with MilliQ water, callose deposition was examined by measuring aniline blue fluorescence intensity with a confocal microscope (405 nm excitation laser, 475–525 nm emission) as described previously (Rinne et al, 2005). Aniline blue quantification was carried out using ImageJ; data was generated by quantifying 50–60 cells per genotype.

## PAC treatment

For paclobutrazol (PAC) treatment, 11WSD plants were moved to LD (with and without cold treatment) and sprayed twice a week with 100 μM PAC (MedChemExpress, USA) in aqueous solution for 4 weeks.

## ChIP assays

Chromatin immunoprecipitation (ChIP) assays were performed as described previously (Gendrel et al, 2005; Saleh et al, 2008), with some modifications. Apices from actively growing hybrid aspen plants were collected and immersed in cross-linking buffer containing 1% formaldehyde, subjected to a vacuum for 20 min, and then glycine added to a final concentration of 0.125 M to stop the cross-linking process. Cross-linked samples were rinsed 3–4 times with water, followed by freezing in liquid nitrogen, and stored at −80 °C. Approximately 1 g of tissue samples was ground into a fine powder and suspended in precooled nuclei isolation buffer. The homogenized mixtures were then gently vortex-mixed and filtered through two layers of Miracloth. The filtered samples were centrifuged, and the pellets were resuspended in nuclei lysis buffer. Chromatin was sonicated using a Bioruptor UCD-300 (Diagenode) to achieve fragments of approximately 0.3–0.5 kb. Following sonication, samples were centrifuged again to remove cell debris, and each supernatant was transferred to a new tube after retaining 10% of each sonicated sample used as the input DNA control in the qPCR analyses. The supernatant was precleared with 40 μl protein A-magnetic beads (Dynabeads, Invitrogen) for 60 min at 4 °C, with gentle agitation and shaking. Fifteen micrograms of anti-Myc monoclonal antibody (Abcam, Cambridge, UK; Cat no. ab32; GR255064) were added to each supernatant, and the resulting mixtures were incubated overnight at 4 °C. Protein A-magnetic beads were added again, and the incubation continued for an additional 2 h. The magnetic beads underwent two washes each with a low salt buffer, high salt buffer, LiCl buffer and TE buffer. Immunocomplexes were collected from the beads with 250 μl of elution buffer, and incubated at 65 °C for 20 min with agitation. NaCl (0.3 M) was added to each tube (including the input DNA control), and cross-linking reversed by overnight incubation at 65 °C. Residual protein was degraded by incubation with 20 mg of proteinase K in 10 mM EDTA and 40 mM Tris-HCl, pH 8.0, at 45 °C for 1 h. After proteinase treatment, precipitated DNA was purified using a ChIP DNA clean and concentrator kit following the manufacturer's protocol (Zymo Research Corp.). Both immunoprecipitated and input DNA were analyzed by real-time PCR using a Light Cycler instrument (CFX96 Real-Time PCR System).

## Yeast one-hybrid

A yeast one-hybrid experiment was conducted as per (Reece-Hoyes and Walhout, 2018). To create the bait construct, the promoter sequence of *GA20ox* was amplified and cloned into the pDONR P4-P1R entry vector. The promoter sequence was transferred to destination vectors PMW2, resulting in promoterGA20ox::HIS3. The full-length CDS of *LIM1* was cloned into the pDEST22 vector to generate the prey construct. The bait and prey constructs were then transformed into yeast strains YM4271 and Yα1867α, respectively. The positive bait transformants were plated on SC-HIS-URA with varying doses of 3AT to perform an auto-activation test. Transformants with minimum autoactivation were selected for mating with a yeast strain that had the prey plasmid. Positive interactions were examined by plating on SC-HIS-URA-TRP with various concentrations of 3AT. The pDEST22-AD empty vector was used as a negative control.

## Modeling

The mathematical model of the signaling network (Fig. 7A) comprises five Ordinary Differential Equations (ODE), together with expressions representing how production of LIM1 and SVL vary over the time scale of weeks after the onset of LT. Full details of the model equations, definitions of model variables and parameters, and strategies used for parameter estimation and model simulation are provided as Appendix Methods. Figure 7B and Appendix Fig. S6A,B present simulations of the model equations with the estimated parameter values listed in Appendix Table S1.

# Data availability

This study includes no data deposited in external repositories.

The source data of this paper are collected in the following database record: biostudies:S-SCDT-10_1038-S44318-024-00256-5.

# Peer review information

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

## Acknowledgements

This project has received funding from the European Union's Horizon 2020 research and innovation program under the Marie Sklodowska-Curie grant agreement No 890883 to SKP. SKP was supported by Marie Skłodowska-Curie Individual Fellowships (DECORE: SEP-210709191), Wenner-Gren fellowship (UPD2019-0203), and Human Frontier Research program (project RGP0002/2020). JPM was supported by the "Seed Grant" money under Dev. Scheme (6031) of Institutions of Eminence (IoE) Scheme of Banaras Hindu University. BA and AN were supported by Human Frontier Research program (project RGP0002/2020). EMB and TSM were supported by the Human Frontier Research program (project RGP0002/2020) and the French government in the framework of the IdEX Bordeaux University "Investments for the Future" program/GPR Bordeaux Plant Sciences (EMB). GWB was supported by Leverhulme grant no. RPG-2019-267, BBSRC grant nos. BB/N009754/1 and BB/S002804/1, and Human Frontiers in Science research grant RGP0002/2020. LB was supported by Human Frontier Science Program grant number RGY0075/2020. XW was supported by the China Scholarship Council under the Grant CSC NO. 202206320357. YM was supported by Shandong Provincial Natural Science Foundation (ZR2021QC058). RPB was supported by Human Frontier Research program (project RGP0002/2020), Knut and Alice Wallenberg Foundation grant (2023.0209) and Vetenskapsrådet (2020-03522).

## Author contributions

**Shashank K Pandey**: Conceptualization; Resources; Data curation; Software; Formal analysis; Funding acquisition; Validation; Investigation; Visualization; Methodology; Writing—original draft; Writing—review and editing. **Jay Prakash Maurya**: Conceptualization; Data curation; Formal analysis; Funding acquisition; Validation; Investigation; Visualization; Methodology; Writing—original draft; Writing—review and editing. **Bibek Aryal**: Data curation; Formal analysis; Validation; Investigation; Visualization; Methodology; Writing—review and editing. **Kamil Drynda**: Conceptualization; Data curation; Software; Formal analysis; Validation; Investigation; Visualization; Methodology; Writing—review and editing. **Aswin Nair**: Data curation; Formal analysis; Validation; Investigation; Visualization; Methodology; Writing—review and editing. **Pal Miskolczi**: Data curation; Formal analysis; Validation; Investigation; Visualization; Methodology; Writing—review and editing. **Rajesh Kumar Singh**: Data curation; Formal analysis; Validation; Visualization; Methodology; Writing—review and editing. **Xiaobin Wang**: Data curation; Formal analysis; Validation; Investigation; Visualization; Methodology. **Yujiao Ma**: Data curation; Formal analysis; Validation; Investigation; Visualization; Methodology. **Tatiana de Souza Moraes**: Methodology; Writing—review and editing. **Emmanuelle M Bayer**: Conceptualization; Funding acquisition; Writing—review and editing. **Etienne Farcot**: Conceptualization; Resources; Data curation; Software; Formal analysis; Supervision; Funding acquisition; Validation; Investigation; Visualization; Methodology; Writing—review and editing. **George W Bassel**: Conceptualization; Funding acquisition; Writing—review and editing. **Leah R Band**: Conceptualization; Resources; Data curation; Software; Formal analysis; Supervision; Funding acquisition; Validation; Investigation; Visualization; Methodology; Writing—original draft; Writing—review and editing. **Rishikesh P Bhalerao**: Conceptualization; Resources; Data curation; Formal analysis; Supervision; Funding acquisition; Validation; Investigation; Visualization; Methodology; Writing—original draft; Project administration; Writing—review and editing.

Source data underlying figure panels in this paper may have individual authorship assigned. Where available, figure panel/source data authorship is listed in the following database record: biostudies:S-SCDT-10_1038-S44318-024-00256-5.

## Funding

## Disclosure and competing interests statement

The authors declare no competing interests.

# Expanded View Figures

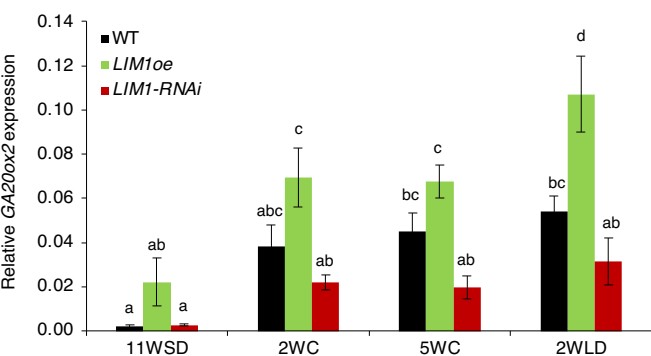

**Figure EV1.  LIM1 is a positive regulator of gibberellic acid (GA) pathway.**

Relative expression of *GA20ox2* in buds of wild-type (WT), *LIM1oe,* and *LIM1-RNAi* plants at 11 weeks short days (11WSD), 2 (2WC), and 5 weeks of cold (5WC) and after 2 weeks of warm temperature (20 °C) in long days (2WLD). The expression values are relative to the reference gene UBQ and the average of three biological replicates. Error bars indicate standard error mean (±SEM). Different letters over the bars indicate statistically significant differences at *P* < 0.05 by one-way ANOVA Duncan's test.

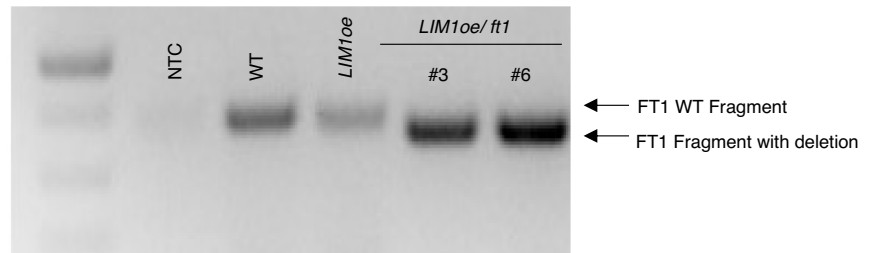

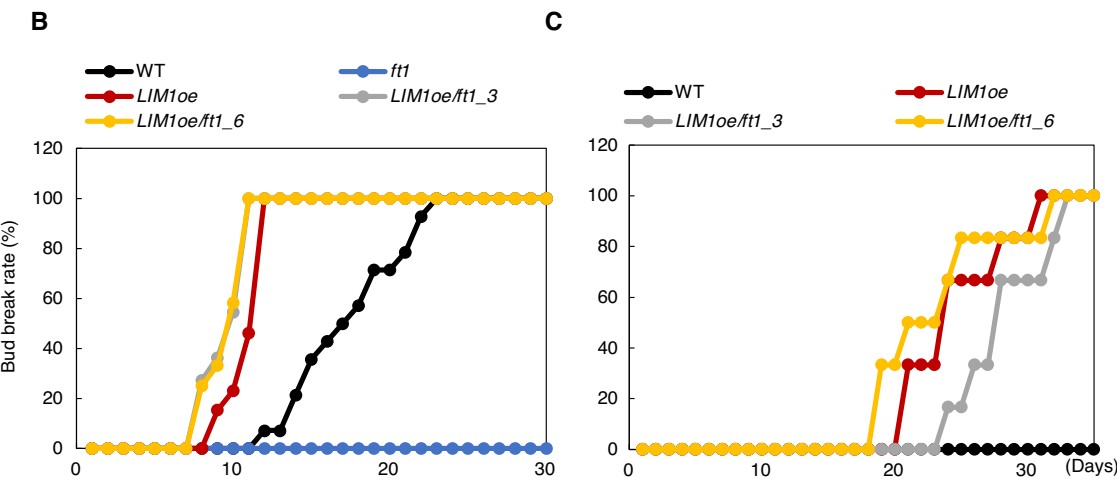

**Figure EV2. LIM1 and FT1 function in a partially redundant manner.**

(A) Screening of *ft1* knockout in *LIM1oe/ft1* double transgenic lines. The polymerase chain reaction (PCR) result showing the detection of CRISPR/Cas9-mediated FT1 deletion in *LIM1oe/ft1* lines by agarose gel electrophoresis. PCR products *c.* 400 bp size corresponded to the wild-type (WT) variant, and the FT1 amplicon in *LIM1oe/ft1* lines of *c.* 350 bp indicated the internal deletion. NTC stands for no template control (B) The bud break rate (%) of WT, *LIM1oe*, and *LIM1oe/ft1* (lines 3 and 6) plants grown under SD for 11 weeks, then treated with cold (4 °C) for 5 weeks followed by transfer to LD for bud burst analysis. (C) The bud break rate (%) of WT, *LIM1oe*, and *LIM1oe/ ft1* plants moved directly from 11 weeks of short-day conditions (SD) to long-day conditions (LD), without a cold treatment, corresponds to Fig. 6A). The experiments (B and C) were repeated at least twice with similar results, and the bud-break rate (%) is shown with data from 7 to 10 plants from each line.

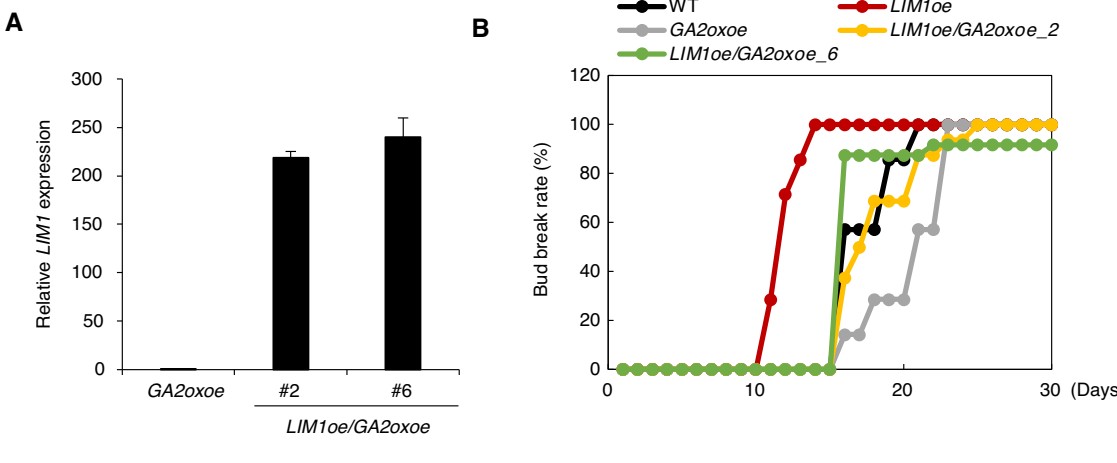

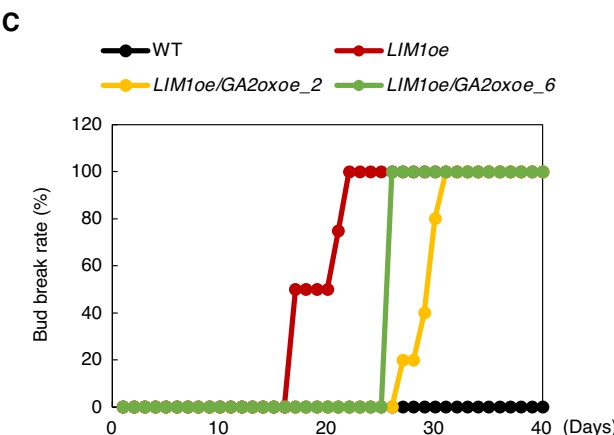

**Figure EV3.   LIM1 and FT1 redundantly converge on the gibberellic acid (GA) pathway.**

(**A**) *LIM1* expression in *LIM1oe/GA2oxoe* lines. See Fig. 6 (**B**). Expression values shown are normalized to the reference gene UBQ and are averages of three biological replicates (±SEM). (**B**) Bud break phenotypes of WT, *LIM1oe, GA2oxoe* and *LIM1oe/GA2oxoe* plants grown under SD for 11 weeks, then treated with cold (4 °C) for 5 weeks followed by transfer to LD for bud burst analysis. (**C**) Bud-break rate (%) of wild-type (WT), *LIM1oe* and *LIM1oe/GA2oxoe* line plants moved directly from 11 weeks of short-day conditions (SD) to long-day conditions (LD) without cold treatment, corresponding to Fig. 6B. The experiments (**B** and **C**) were repeated at least twice with similar results, and the bud-break rate (%) is shown with data from 7 to 10 plants from each line.

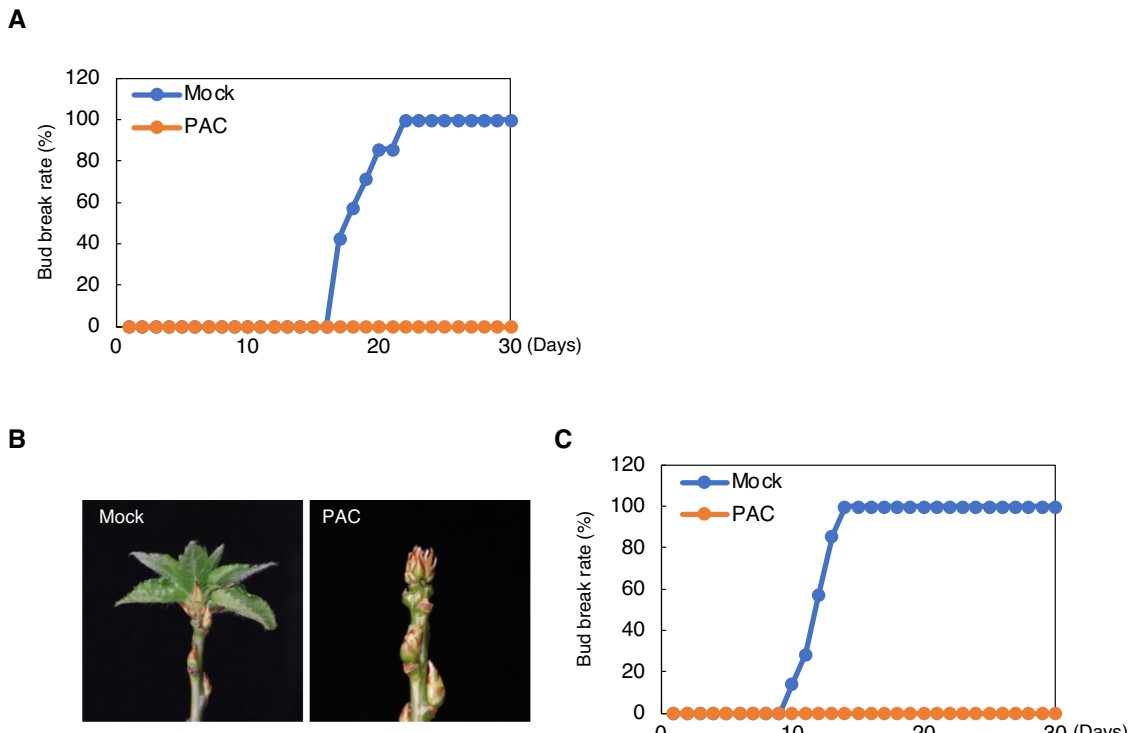

**Figure EV4. GA is the downstream target of *LIM1* and *FT1*.**

(A) Bud-break rate (%) of *LIM1oe/FT1crispr* plants treated with Mock and paclobutrazol (PAC) moved directly from 11 weeks of short-day (SD) to long-day conditions (LD) without cold treatment, corresponding to Fig. 6C. (B, C) Bud break phenotypes of *LIM1oe/FT1crispr* plants treated with Mock and paclobutrazol (PAC) under long-day conditions (LD) after cold treatment. Bud break phenotyping were repeated at least twice with similar results, and the bud-break rate (%) is shown with data from 7 plants.

