## [Peer Review File · The EMBO Journal]

A regulatory module mediating temperature control of cell-cell communication facilitates tree bud dormancy release

Rishikesh Bhalerao, Shashank Pandey, Jay Maurya, Bibek Aryal, Kamil Drynda, Aswin Nair, Pal Miskolczi, Rajesh Singh, Xiaobin Wang, Yujiao Ma, Tatiana Souza-Moraes, Emmanuelle Bayer, Etienne Farcot, George Bassel, and Leah Band

Corresponding author(s): Rishikesh Bhalerao (Rishi.Bhalerao@slu.se)

Review Timeline:

Submission Date:	27th Mar 24
Editorial Decision:	3rd May 24
Revision Received:	26th Jun 24
Editorial Decision:	9th Aug 24
Revision Received:	21st Aug 24
Accepted:	12th Sep 24

Editor: William Teale

Transaction Report:

Dear Rishi,

Thank you again for the submission of your manuscript entitled "A regulatory module mediating temperature control of cell-cell communication facilitates bud dormancy release" (EMBOJ-2024-117420) and for your patience during the review process. Your manuscript was sent for appraisal to two referees; we have now received the reports from both of them, which I copy below.

As you can see from their comments, both appreciated the approach you took and saw potential value in your work. However, there is a substantial list of issues that need to be addressed, some of which will require a return to the lab. In principle, The EMBO Journal is interested in the manuscript; however, we cannot proceed towards publication without a substantial round of revision.

Before inviting you to address the comments of all referees in a revised version of the manuscript, therefore, I suggest we meet via Zoom (once you have had a chance to digest the reports) to discuss the best way forward. I understand the extra difficulties that working with aspen presents; we have to work out together whether satisfying the referees is realistic within a reasonable time-frame, or whether another option might be more prudent. The EMBO Journal policy can only allow only a single major round of revision and that it is therefore important to resolve the main concerns at this stage.

I would also like to point out that as a matter of policy, competing manuscripts published during this period will not be taken into consideration in our assessment of the novelty presented by your study ("scooping" protection). We have extended this 'scooping protection policy' beyond the usual 3 month revision timeline to cover the period required for a full revision to address the essential experimental issues. Please contact me if you see a paper with related content published elsewhere to discuss the appropriate course of action.

Again, please contact me at any time during revision if you need any help or have further questions.

Thank you very much again for the opportunity to consider your work for publication. I look forward to your revision.

Best regards,

William

William Teale, Ph.D.
Editor
The EMBO Journal

When submitting your revised manuscript, please carefully review the instructions below and include the following items:

- 1) a .docx formatted version of the manuscript text (including legends for main figures, EV figures and tables). Please make sure that the changes are highlighted to be clearly visible.
- 2) individual production quality figure files as .eps, .tif, .jpg (one file per figure).
- 3) a .docx formatted letter INCLUDING the reviewers' reports and your detailed point-by-point response to their comments. As part of the EMBO Press transparent editorial process, the point-by-point response is part of the Review Process File (RPF), which will be published alongside your paper.
- 4) a complete author checklist, which you can download from our author guidelines ([https://wol-prod-cdn.literatumonline.com/pb-assets/embo-site/Author Checklist%20-%20EMBO%20J-1561436015657.xlsx](https://wol-prod-cdn.literatumonline.com/pb-assets/embo-site/Author%20Checklist%20-%20EMBO%20J-1561436015657.xlsx)). Please insert information in the checklist that is also reflected in the manuscript. The completed author checklist will also be part of the RPF.
- 5) Please note that all corresponding authors are required to supply an ORCID ID for their name upon submission of a revised manuscript.
- 6) We require a 'Data Availability' section after the Materials and Methods. Before submitting your revision, primary datasets produced in this study need to be deposited in an appropriate public database, and the accession numbers and database listed under 'Data Availability'. Please remember to provide a reviewer password if the datasets are not yet public (see

<https://www.embopress.org/page/journal/14602075/authorguide#datadeposition>). If no data deposition in external databases is needed for this paper, please then state in this section: This study includes no data deposited in external repositories. Note that the Data Availability Section is restricted to new primary data that are part of this study.

Note - All links should resolve to a page where the data can be accessed.

8) For data quantification: please specify the name of the statistical test used to generate error bars and P values, the number (n) of independent experiments (specify technical or biological replicates) underlying each data point and the test used to calculate p-values in each figure legend. The figure legends should contain a basic description of n, P and the test applied. Graphs must include a description of the bars and the error bars (s.d., s.e.m.).

9) We would also encourage you to include the source data for figure panels that show essential data. Numerical data can be provided as individual .xls or .csv files (including a tab describing the data). For 'blots' or microscopy, uncropped images should be submitted (using a zip archive or a single pdf per main figure if multiple images need to be supplied for one panel). Additional information on source data and instruction on how to label the files are available at .

10) We replaced Supplementary Information with Expanded View (EV) Figures and Tables that are collapsible/expandable online (see examples in <https://www.embopress.org/doi/10.15252/embj.201695874>). A maximum of 5 EV Figures can be typeset. EV Figures should be cited as 'Figure EV1, Figure EV2" etc. in the text and their respective legends should be included in the main text after the legends of regular figures.

12) Our journal encourages inclusion of *data citations in the reference list* to directly cite datasets that were re-used and obtained from public databases. Data citations in the article text are distinct from normal bibliographical citations and should directly link to the database records from which the data can be accessed. In the main text, data citations are formatted as follows: "Data ref: Smith et al, 2001" or "Data ref: NCBI Sequence Read Archive PRJNA342805, 2017". In the Reference list, data citations must be labeled with "[DATASET]". A data reference must provide the database name, accession number/identifiers and a resolvable link to the landing page from which the data can be accessed at the end of the reference. Further instructions are available at .

Further instructions for preparing your revised manuscript:

We realize that it is difficult to revise to a specific deadline. In the interest of protecting the conceptual advance provided by the work, we recommend a revision within 3 months (1st Aug 2024). Please discuss the revision progress ahead of this time with the editor if you require more time to complete the revisions. Use the link below to submit your revision:

Referee #1:

Pandey et al. report in this manuscript on the identification of poplar Low temperature induced MADS-box1 (LIM1) a previously uncharacterized member of the MADS box gene family that is here shown to regulate bud dormancy release, by acting as direct upstream activator of the GA pathway. LIM1 is observed to directly activate GA20ox in addition to FT1, concurrent GA pathway and FT1 activity appearing to be essential for bud activation.

Low temperature induces LIM1 expression in dormant buds, correlating with opening of PD, whereas this activation is independent of DAM/SVLs or FT1. Bud break is actually delayed in LIM1-RNAi while accelerated in LIM1oe lines, these last initiating bud break even in the absence of a 5WC treatment. Authors show that callose levels are reduced in LIM1oe buds and elevated in RNAi lines, LIM1oe plants also initiating bud break on grafting FT1oe stocks, in line with LIM1 acting as a negative regulator of callose accumulation and therefore promoting PD opening. FT1 and GA20-oxidase gene expression is on the other side induced in LIM1oe lines and reduced in RNAi lines, LIM1 being shown by yeast one-hybrid and ChIP-PCR studies to directly bind the promoter of the GA20 oxidase gene. Callose levels are on the other side shown to be reduced in GA20ox-oe lines while elevated in lines overexpressing GA2 oxidase, in support of GA exerting a negative role of on callose accumulation. LIM1, FT1 and GA20 oxidase are all induced in response to low temperature, and notably, whereas FT1 inactivation does not impair LIM1oe earlier bud break, bud outgrowth was observed to be delayed in LIM1oe/GA20ox-lines. LIM1oe/ft1 bud break is moreover blocked by PAC, in support of LIM1 bypassing need of FT1, by directly activating the GA pathway. Mathematical modelling of LIM1 promotion of FT expression and the upregulation of GA20 oxidase by both LIM1 and FT, showed this interaction to be sufficient to explain bud break and gene expression dynamics. However, the model requires increase of LIM1 to be gradual which can be attained by introducing a negative GA feed-back on LIM1 and in agreement with this regulation LIM1 expression level are notably reduced in GA20ox-oe lines.

Overall, these are highly significant and novel results identifying a central role of LIM1 in cold dormancy release and a convergent function of this factor and FT1 on GA biosynthesis and PD opening, besides providing important insights on the topology of this regulatory network.

There are however a few issues that might need to be addressed for publication:

- 1) While GA20ox is increased in LIM1oe and reduced in LIM1-RNAi lines as expected for direct LIM1 regulation of this gene, behavior of the FT1 target in Figure 4 is notably different: FT1 is strongly upregulated in LIM1oe lines at 11WSD, 2WC and 5WC, but drastically suppressed after 2WLD. This appears to indicate a switch in LIM1 transcriptional activity between SD and LD or in response to increased GA levels which is not contemplated in the model. This regulation is moreover dependent on LIM1 since it is not observed in WT plants.
- 2) Callose deposition is shown on Figure 5 to be reduced in GA20ox lines and increased in GA2ox lines. However, callose levels are also slightly different in the WT between both experiments, which contributes to make differences in transgenic lines

statistically significant. Reduction of callose after 5WC is also more evident in the GA20ox than in the GA2ox experiment, indicating that these differences vary between experiments. Providing some of the images used to these measurements would help readers to get an idea on how robust this response is.

3) Earlier bud break of LIM1oe lines is not affected by CRISPR knockout of FT1 but is delayed in LIM1oe/GA2ox-oe lines. PAC on the other side suppresses LIM1oe/ft1 bud break on direct transfer to LD conditions, indicating that both lack of FT1 and GA is needed to suppress bud growth phenotype of LIM1oe lines as independent of low temperature dormancy release. Here it would be relevant analyzing FT1 expression levels both in LIM1oe and LIM1oe/ft1 lines.

4) Low temperatures up-regulate in the proposed model FT1 expression by suppressing its negative regulation by SVPs, besides promoting FT1 activation by LIM1. However, based on Figure 1, LIM1 might undergo in LD a switch in its transcriptional activity, as indicated by the strong suppression of FT1 expression in LIM1oe lines. This regulation does not seem to be considered in the model and would be critical to a complete understanding of this network. DELLAs are known to directly interact with the MADS box factors and direct LIM1-DELLAs interaction might for instance mediate this transcriptional switch. Analyzing FT1 expression levels in LIM1oe/GA2ox, and GA2ox lines would be informative into this respect.

5) Modeling showed slow LIM1 activation to be essential for observed bud growth phenotypes. Authors favor a model where negative feed-back GA regulation of LIM1 exerts this control and indeed LIM1 expression is shown in Figure 7 to be reduced in GA20ox lines. However, it cannot be excluded that epigenetics regulation of LIM1 plays also a relevant role.

6) Number of plant replicates used to analyze bud break in Figures S6 and S7 should be specified.

7) Without 5WC treatment LIM1oe/GA2ox lines still undergo bud break, but this is delayed with respect to the LIM1oe lines. The model predicts GA20ox and GA levels to be elevated in LIM1oe lines while reduced in RNAi, consistently with GA20ox expression levels observed in these plants. However, it fails to correctly predict 5WC FT1 expression levels in LIM1oe lines. Is there any possible explanation to this deviation? Would it be possible that LIM1 regulates its own expression?

Referee #2:

In this manuscript, the authors focused on low-temperature (LT)-induced cell-cell communication via plasmodesmata in regulating bud dormancy in trees. They identified a MADS-box factor, LIM1 as a low-temperature-induced activator of GA pathway in hybrid aspen. The authors made the following claims. LT activates the expression of LIM1 in dormant buds and LIM1 mediates dormancy release and promotes bud break. LIM1 suppresses callose accumulation and mediates PD opening. LIM1 is a positive regulator of FT1 and the GA pathway, which negatively regulates callose accumulation. LIM1 and FT1 act along partially redundant pathways in buds. Their mathematical modelling and experimental validation demonstrate a negative feedback regulation of LIM1 by GA. Overall, the data are not solid enough to support their claims.

Major comments:

Based on the result of Figure 1, the authors stated that LT activates LIM1 expression in dormant buds. It may be necessary to include a positive control (a previously characterized gene whose expression is induced by LT in dormant buds) and a negative control (not induced by LT) in the expression assay.

Figure S1, the sequence itself of LIM1 is meaningless. Provide a protein structural diagram. It is highly recommended to perform a phylogenetic tree analysis for LIM1 and its orthologs during evolution. I am also wondering whether the expression of its most closed homolog gene(s) in hybrid aspen is induced by LT.

The authors stated that "LIM1 suppresses callose accumulation and mediates PD opening", and PD is a keyword of this study. While they used callose levels and bud reactivation assay to reflect the status of PD, direct evidence of PD opening/closure is still lacking. It is highly recommended to provide cell biological evidence (such as fluorescent markers) for PD status.

The authors concluded that "Compared with WT buds, LT activation of FT1 was higher in LIM1oe, and LIM1 downregulation resulted in reduced FT1 induction by LT (Fig. 4A)", which was not fully supported by the data shown in Figure 4A. It is quite surprising why there was no significant difference of FT1 expression between LIM1-RNAi and WT. Please explain.

Since LIM1 regulates both FT1 and GA20ox expression, and the authors showed that LIM1 binds to GA20ox (Figure 4C, D), I am wondering whether LIM1 directly binds to the promoter sequence of FT1. I suggest the authors to perform Y2H and ChIP assays for FT1. Moreover, a negative control is required in the ChIP-PCR assay, e.g. PCR of an unrelated gene or the coding region of FT1. In addition, EMSA would be an ideal and direct experiment to support DNA-protein interaction in vitro.

A follow-up question, how many genes encode GA20-oxidase in hybrid aspen? If more than two genes of GA20ox, which gene was analyzed in this study?

The authors concluded that LIM1 and FT1 function in a partially redundant manner that converges on the gibberellic acid (GA) pathway based on the bud-break phenotypes in Figure 7. Quantification data of the bud-break phenotype and their statistical analysis are required.

At the beginning of the study, the authors screened gene expression data for transcription factors that were induced by LT. Then

how many transcription factors in total were analyzed and how many of them showed increased expression? Can you briefly tell us the results?

"Altogether these results suggest that LIM1 is not a downstream target of SVL or FT1 pathway in LT response and distinct pathways MAY BE INVOLVED IN regulating repression of SVL and induction of LIM1 by LT."

The data of Figure S4 could be combined to Figure 2.

Dear Editor,

Please find enclosed our revised manuscript EMBOJ-2024-117420. We have addressed the issues raised by the reviewers point by point to the best of our abilities. Please do not hesitate to contact us if additional information is needed.

Sincerely,

Rishikesh P. Bhalerao

Response to reviewers:

Referee #1:

1) While GA20ox is increased in LIM1^{oe} and reduced in LIM1-RNAi lines as expected for direct LIM1 regulation of this gene, behavior of the FT1 target in Figure 4 is notably different: FT1 is strongly upregulated in LIM1^{oe} lines at 11WSD, 2WC and 5WC, but drastically suppressed after 2WLD. This appears to indicate a switch in LIM1 transcriptional activity between SD and LD or in response to increased GA levels which is not contemplated in the model. This regulation is moreover dependent on LIM1 since it is not observed in WT plants.

Re: We appreciate the reviewer's comment. It is important to note however that as bud break proceeds, FT1 expression is rapidly downregulated especially at later stage of bud break as shown earlier (Andre et al., 2022) as well as below (Fig. 1). We have analysed FT1 expression at 2 week LD which is very early stage of bud break when FT1 is not yet downregulated in the wild type. As LIM1^{oe} is very early in bud break, LIM1^{oe} buds at 2 WLD are far more advanced in bud break compared to wild type and this presumably explains low FT1 expression at 2WLD in LIM1^{oe} buds compared to WT.

Fig. 1 (A) Expression of FT1 in the wild type buds during bud break.

Fig. 1 (B) Advanced bud break in LIM1oe compared to wild type (T89).

It is of course possible that there is a switch in transcriptional activity of LIM1 between SD and LD that is GA dependent and we have addressed this experimentally also in response to the later comment (4). However, the point is well taken and we have added this to discussion as follows:

Intriguingly, in contrast to positive effect of LIM1oe on FT1 expression in LT treated buds, FT1 expression was significantly reduced in LIM1oe after transfer from LT to warm temperature. It has been shown that FT1 expression is rapidly downregulated in late stages of bud break (Andre et al., 2022) and since bud break is significantly advanced in LIM1oe (compared to wild type, Fig. 4A), FT1 downregulation may reflect advanced bud break in LIM1oe compared to wild type. Additionally, it is also possible that there is a switch in LIM1 activity from activator to repressor during bud break which may contribute to repression of FT1 in LIM1oe. The MADS-box transcription factors are known to heterodimerize which can modulate the properties of resulting complexes (Puranik *et al.*, 2014). Interestingly, in apple buds, diverse MADS-box complexes are associated with distinct stages of dormancy and bud break (da Silveira Falavigna *et al.*, 2021). Thus, it is possible that LIM1 may interact with MADS-box proteins during bud break that are distinct from those during dormancy release which could contribute to such a switch in LIM1 activity during bud break.

2) Callose deposition is shown on Figure 5 to be reduced in GA20ox lines and increased in GA2ox lines. However, callose levels are also slightly different in the WT between both experiments, which contributes to make differences in transgenic lines statistically significant. Reduction of callose after 5WC is also more evident in the GA20ox than in the GA2ox experiment, indicating that these differences vary between experiments. Providing some of the images used to these measurements would help readers to get an idea on how robust this response is.

Re: Thanks for raising this. Care was taken to use the same settings within experiments but as the WT vs GA20ox and WT vs GA2ox are independent experiments performed at different time points and thus wild types in these experiments are not directly comparable to each other.

As high GA levels (in GA20ox) downregulate callose and conversely, low GA (in GA2ox) enhance callose levels, this leads to reduction of callose at 5WC being higher in GA20ox than in GA2ox. We have now provided representative images (Fig. 2) as suggested (see below). We apologise as the images do not transfer well to a word document unlike on screen and it should be noted that simple visual analysis with representative images do not always capture

quantitative differences that are revealed by the quantification shown in the graphs that are based on several measurements of multiple cells.

3) Earlier bud break of LIM1oe lines is not affected by CRISPR knockout of FT1 but is delayed in LIM1oe/GA2ox-oe lines. PAC on the other side suppresses LIM1oe/ft1 bud break on direct transfer to LD conditions, indicating that both lack of FT1 and GA is needed to suppress bud growth phenotype of LIM1oe lines as independent of low temperature dormancy release. Here it would be relevant analyzing FT1 expression levels both in LIM1oe and LIM1oe/ft1 lines.

Re: Thanks for pointing this. We have already reported on FT1 expression in LIM1oe (Fig. 4A in the manuscript) and also show here FT1 in LIM1oe/ft1 (Fig. 3). As can be seen there is transcript of FT1 in LIM1oe/ft1 buds. However since ft1 allele carries an internal deletion, this is nonfunctional.

Fig. 3 FT1 expression in wild type (T89), and LIM1oe/ft1 mutant. It should be noted that ft1 mutant allele carries a deletion and is therefore non-functional.

However, to clarify, for bud break, dormancy release is completely essential. When LIM1 is overexpressed, dormancy is defective and loss of FT1 (in LIM1oe/ft1) does not suppress this dormancy defect and bud break is not affected. However when GA is reduced by overexpression of GA2-oxidase in LIM1oe, bud break is delayed as GA is crucial for bud break post-dormancy release as shown in earlier studies (Singh et al., 2018). However when both FT1 is deleted and GA levels are reduced simultaneously in LIM1oe (i. e. PAC treated

LIM1oe/ft1), dormancy release is completely blocked as shown by lack of response of buds to cold and consequently bud break never takes. Thus in this case (PAC treated LIM1oe/ft1), failure to undergo bud break is a consequence of failure of LIM1oe/ft1 to undergo dormancy release (rather than FT1 and GA acting together in bud break after dormancy release). We have emphasized this more clearly now.

We have now added the following sentence to clarify in results section: **The complete blockage of bud break in paclobutrazol treated LIM1oe/ft1 (in contrast to delay in bud break in LIM1oe/GA2oxoe) even after cold treatment indicates that blocking both FT1 and GA pathway simultaneously, prevents the release of dormancy itself, which then is reflected in a failure to undergo bud break in these plants.**

4) Low temperatures up-regulate in the proposed model FT1 expression by suppressing its negative regulation by SVPs, besides promoting FT1 activation by LIM1. However, based on Figure 1, LIM1 might undergo in LD a switch in its transcriptional activity, as indicated by the strong suppression of FT1 expression in LIM1oe lines. This regulation does not seem to be considered in the model and would be critical to a complete understanding of this network. DELLAs are known to directly interact with the MADS box factors and direct LIM1-DELLAs interaction might for instance mediate this transcriptional switch. Analyzing FT1 expression levels in LIM1oe/GA2ox, and GA2ox lines would be informative into this respect.

Re: Thank you for raising the intriguing possibility of LIM1-DELLA interaction and its possible role in the switch of LIM1 by GA. Indeed the effect of change in GA levels could be mediated via LIM1 interaction with DELLA proteins. To test this interesting possibility, we therefore expressed RFP-LIM1 and RGA-Myc in protoplast and then used the isolated proteins to test whether these proteins interact. But as can be seen, we did not find such an interaction (Fig. 4). Thus the potential switch in LIM1 could be via some other mechanism. For example, MADS box proteins can heterodimerize and this modulates their properties and thus interaction of LIM1 with other bud break specific MADS box factors could contribute to a potential switch in LIM1. We have discussed this possibility as well.

Fig. 4 Expression of RFP-LIM1 and RGA-Myc in protoplast followed by co-IP.

Moreover, as also outlined in response to point 1, FT1 is downregulated as bud break proceeds in the wild type and since bud break in LIM1^{oe} is much more rapid and advanced, the reduction in FT1 in LIM1^{oe} could also represent the difference in stages of bud break between wild type (early) and LIM1^{oe} (late).

Additionally, as LIM1 does not bind to FT1 promoter, this regulation may be more complex via some other targets that are currently unknown. Thus while intriguing, we feel it is out of scope of this current study. However as also in reply to point 1, we have discussed the downregulation raised by the reviewer of a switch in LIM1 in discussion. We hope this is acceptable.

5) Modeling showed slow LIM1 activation to be essential for observed bud growth phenotypes. Authors favor a model where negative feed-back GA regulation of LIM1 exerts this control and indeed LIM1 expression is shown in Figure 7 to be reduced in GA20ox lines. However, it cannot be excluded that epigenetics regulation of LIM1 plays also a relevant role.

Re: We thank the reviewer for pointing this out. We agree and have now included this in discussion as follows:

While we find evidence for negative feedback regulation of *LIM1*, additional factors e. g. epigenetic regulation may also contribute to precise *LIM1* response to LT as has been observed in case of LT regulation of MADS-box factors such as *FLC-like* gene in axillary buds of Kiwi or DAM genes in buds of various tree species (Wu et al., 2017; Voogd et al, 2022; Sato and Yamane, 2024).

6) Number of plant replicates used to analyze bud break in Figures S6 and S7 should be specified.

Re: this is now indicated.

7) Without 5WC treatment LIM1^{oe}/GA20ox lines still undergo bud break, but this is delayed with respect to the LIM1^{oe} lines. The model predicts GA20ox and GA levels to be elevated in LIM1^{oe} lines while reduced in RNAi, consistently with GA20ox expression levels observed in these plants. However, it fails to correctly predict 5WC FT1 expression levels in LIM1^{oe} lines. Is there any possible explanation to this deviation? Would it be possible that LIM1 regulates its own expression?

Re: Thanks for pointing this out. In addition to LIM1, epigenetics (as pointed by the reviewer above), could contribute to FT1 regulation which are not considered in the model and this could be the reason why the model fails to fully capture FT1 regulation. We have now added this in the results and discussion as follows:

One exception to this was FT1 upregulation in LIM1^{oe} which could hint at additional mechanisms such as epigenetic regulation that have been shown to mediate in LT response of gene expression in buds (Sato & Yamane, 2024) could also contribute to FT1 regulation which is not captured in the modelling.

We have now also performed analysis of LIM1 expression (Fig. 5) to check autoregulation as suggested by the reviewer by checking expression of native LIM1 (using specific primers that detect endogenous LIM1 only) in LIM1oe and do not find evidence of autoregulation.

Fig. 5 Expression of endogenous LIM1 in LIM1oe compared with the wild type (T89).

Referee #2:

In this manuscript, the authors focused on low-temperature (LT)-induced cell-cell communication via plasmodesmata in regulating bud dormancy in trees. They identified a MADS-box factor, LIM1 as a low-temperature-induced activator of GA pathway in hybrid aspen. The authors made the following claims. LT activates the expression of LIM1 in dormant buds and LIM1 mediates dormancy release and promotes bud break. LIM1 suppresses callose accumulation and mediates PD opening. LIM1 is a positive regulator of FT1 and the GA pathway, which negatively regulates callose accumulation. LIM1 and FT1 act along partially redundant pathways in buds. Their mathematical modelling and experimental validation demonstrate a negative feedback regulation of LIM1 by GA. Overall, the data are not solid enough to support their claims.

Major comments:

Based on the result of Figure 1, the authors stated that LT activates LIM1 expression in dormant buds. It may be necessary to include a positive control (a previously characterized gene whose expression is induced by LT in dormant buds) and a negative control (not induced by LT) in the expression assay.

Re: We have now performed this experiment and data (Fig. 6) has been included as supplementary figure.

Fig. 6 Expression of LT induced gene EBB3 (positive control) and negative control not induced by LT.

Figure S1, the sequence itself of LIM1 is meaningless. Provide a protein structural diagram. It is highly recommended to perform a phylogenetic tree analysis for LIM1 and its orthologs during evolution. I am also wondering whether the expression of its most closed homolog gene(s) in hybrid aspen is induced by LT.

Re: We have now added protein structural diagram (Fig 7A) and added phylogenetic tree as suggested (Fig. 7B).

MVRGKVLQRIEDKSSRQVCFSKRKRGLLKKAKELSVLCDVEMAVIIFSSTGKL
 FEFCSGNSLRNILERYDTHKTKSQEIAICKNVDKTKQNHHAEMYGSSYMDANPL
 QMVQRYFEGKNIEQLNITQLMQLERELDSTLLYTRGRKTEAMMKSVTALHQKEQ
 DLTDENNLIEREISAIINNGNLAGQHGRVVEDPDCVHPSPLDLFHF

Fig. 7A Amino acid sequence of LIM1 with domains of interest (MADS and K-box) indicated.

Fig. 7B Phylogenetic tree with LIM1 and closely related genes indicated in red box.

We have also extracted expressions of closely related genes to LIM1 (in the red box) from RNAseq data (indicated in Table 1) and these are induced also by LT. We feel however that this is outside the scope of the article and thus prefer not to include it in the manuscript unless reviewer feels it is essential.

		log2FoldChange	pvalue	padj
2WC	Potra2n3c6957	0.926678567989601	1.14850773070267e-07	4.32433176669765e-07
	Potra2n1c481	2.72307473076178	4.26928656709607e-131	5.34962953289973e-128
5WC	Potra2n3c6957	0.863941071678713	1.06019593645594e-06	3.16153859632583e-06
	Potra2n1c481	3.30660611044796	7.19859836743271e-194	2.00448970762479e-190

Table 1: Expression extracted from RNAseq data of wild type buds of Potra2n3c6957 and Potra2n1c481 that are closest to LIM1 after 2 and 5 weeks cold.

The authors stated that "LIM1 suppresses callose accumulation and mediates PD opening", and PD is a keyword of this study. While they used callose levels and bud reactivation assay to reflect the status of PD, direct evidence of PD opening/closure is still lacking. It is highly recommended to provide cell biological evidence (such as fluorescent markers) for PD status.

Re: Thanks for raising this issue. To clarify, previous studies (including in tree buds) have established change in callose levels report on PD dynamics with high callose associated with closed and low callose with open PDs (Rinne et al., 1998; Tylewicz et al., 2018). Here we use two independent approaches to assess PD status: First with callose measurements and second by assay of bud break in buds grafted on root stocks of FT1oe plants under non-inductive conditions which reports on PD dependent cell-cell communication, with bud break requiring cell-cell communication via open PDs as shown earlier (Tylewicz et al., 2019). The advantage of the second assay is that FT1 is an endogenous regulator of dormancy /bud break and can only induce bud break under non-inductive conditions only when cell-cell communication is active with open PDs. This is shown by previously published data showing the failure of FT1 to induce bud break in wild type buds with high callose and closed PDs. In contrast, FT1 is able to induce bud break in *abi1-1* mutant buds which were shown to have open PDs (Tylewicz et al., 2018). Thus grafting of buds on FT1oe and assessment of bud break under non-inductive conditions reports whether cell-cell communication is active or blocked and in combination with callose analysis provides insight into open/closed status of PDs and essentially provides the same information as is obtained by use of fluorescent markers that are challenging to use in complex, thick tissue as buds. Thus, we feel that our data using two independent approaches based on callose measurements as well as bud break assay would be consistent with proposed role of LIM1 in low temperature mediated control of PD opening. We hope this addresses the issue raised by the reviewer.

The authors concluded that "Compared with WT buds, LT activation of FT1 was higher in LIM1oe, and LIM1 downregulation resulted in reduced FT1 induction by LT (Fig. 4A)", which was not fully supported by the data shown in Figure 4A. It is quite surprising why there was no significant difference of FT1 expression between LIM1-RNAi and WT. Please explain.

Re: Thanks for pointing this out. We apologise for not stating this clearly. To clarify, FT1 is expressed at extremely low levels even in the wild type buds and thus detecting further reduction in FT1 expression is challenging. Moreover, in LIM1RNAi, the LIM1 expression is not fully suppressed and residual LIM1 might make reduction more difficult to observe. These two factors make it more difficult to detect major reduction in FT1 expression in LIM1RNAi compared high upregulation of FT1 in LIM1oe showing LIM1 effect on FT1 expression. However, despite these challenges, in two out of three biological replicates, FT1 expression is 50% lower in LIM1RNAi than in wild type at 2 weeks cold and 5 weeks cold which is indicative of the overall trend of LIM1 downregulation leading to FT1 downregulation.

		FT1 expression in T89 and LIM1-RNAi			
		10W	2WC	5WC	2WLD
WT	1st	0,00020388	0,01165156	0,04010706	0,09875516

	2nd	0,00023637	0,01263293	0,03459402	0,15247733
	3rd	0,00087810	0,00658470	0,06051067	0,06440575
LIM1-RNAi	1st	0,00021254	0,00343216	0,04338480	0,06999198
	2nd	0,00046194	0,00519018	0,01815696	0,12500000
	3rd	0,00066088	0,00663050	0,02497566	0,12792174

We have now clearly clarified this as follows:

Compared with WT buds, LT activation of FT1 was higher in LIM1oe. Conversely, in two out of three biological replicates, *FT1* expression in LIM1RNAi buds was 50% lower than in the wild type, whereas one replicate showed the same expression as wild type after LT. The less pronounced effect of LIM1 downregulation on *FT1* induction by LT (in contrast with significant upregulation in LIM1oe) could be due to LIM1 being downregulated (but not completely suppressed in LIM1RNAi) with the residual LIM1 may be sufficient to maintain induction of FT1 expression which makes it difficult to detect strong downregulation in FT1. Finally, as FT1 is expressed to very low levels, detection of its downregulation is more challenging (compared to its upregulation). Nevertheless, the overall trend indicates lower FT1 expression in *LIM1* downregulated buds after LT treatment (Fig. 4A).

Since LIM1 regulates both FT1 and GA20ox expression, and the authors showed that LIM1 binds to GA20ox (Figure 4C, D), I am wondering whether LIM1 directly binds to the promoter sequence of FT1. I suggest the authors to perform Y2H and ChIP assays for FT1. Moreover, a negative control is required in the ChIP-PCR assay, e.g. PCR of an unrelated gene or the coding region of FT1. In addition, EMSA would be an ideal and direct experiment to support DNA-protein interaction in vitro.

Re: We thank the reviewer. We have performed ChIP-PCR to investigate if LIM1 binds to FT1 promoter at the MADS-box binding site (Fig. 8). However unlike GA20-oxidase, in which we can clearly demonstrate LIM1 binding at the MADS-box binding site, we do not find any significant binding of LIM1 at FT1 promoter over the MADS-box site. Therefore we conclude that LIM1 is upstream of FT1, but not a direct regulator of FT1. We have now added this data as supplementary figure to the manuscript.

Fig. 8 ChiP-PCR of LIM1-myc binding in FT1 promoter at MADS-box binding site (top panel) and negative control (bottom panel).

A follow-up question, how many genes encode GA20-oxidase in hybrid aspen? If more than two genes of GA20ox, which gene was analyzed in this study?

Re: There is at least 1 more gene and the expression of this gene is analysed (Fig. 9). As all other analysis was performed with the other gene, we suggest that this data could potentially not be included unless reviewer would suggest.

Fig. 9 Expression of GA20ox2 gene homologous to GA20ox analysed in the manuscript.

The authors concluded that LIM1 and FT1 function in a partially redundant manner that converges on the gibberellic acid (GA) pathway based on the bud-break phenotypes in Figure 7. Quantification data of the bud-break phenotype and their statistical analysis are required.

Re: The fig 7 does not include bud break data. Perhaps the reviewer means Fig. S7? If so quantification is now added.

At the beginning of the study, the authors screened gene expression data for transcription factors that were induced by LT. Then how many transcription factors in total were analyzed and how many of them showed increased expression? Can you briefly tell us the results?

Re: We have used the data from Karlberg et al (2010) and subsequent unpublished RNAseq data. In total 88 TFs were upregulated in cold, and of the MADS box related genes, 6 genes were upregulated. MADS-box genes are interesting given their role in dormancy as well as in flowering in which both FT and GA are involved. The selection of LIM1 was based on LIM1 expression but also because LIM1 does not have a closely related paralog which makes it functional analysis more easy compared to making double or multiple knockouts. We have also indicated this the revised manuscript as follows:

To determine the genetic regulators of LT-induced PD opening, we screened gene expression data for transcription factors that were induced in hybrid aspen dormant buds after LT, correlating with opening of PD (Karlberg *et al.*, 2010) and selected transcription factors whose expression matched LT induction of FT1 and GA20-oxidase (a key enzyme in GA biosynthetic pathway), (André *et al.*, 2022; Eriksson *et al.*, 2000; Sheng *et al.*, 2023), the key components implicated in dormancy release and bud break (Singh *et al.*, 2018). This analysis revealed 88 transcription factors induced by LT. Of these LT induced transcription factors, was a previously uncharacterized MADS-box transcription factor, *LIM1* (Fig. S1). *LIM1* encodes a protein with 208 amino acid and has conserved MADS-box and K-box, characteristic of MADS-box transcription factors (Fig. S1A). Transcription factors of MADS-

box family have been implicated in dormancy induction, in several tree species (Singh et al., 2019; Yamane et al., 2019; Moser et al., 2020; Zhao et al., 2023). In contrast, antagonistically acting, promoters of dormancy release are much less studied. Moreover, a large number of genes in *Populus* are often encoded by two closely related paralogs due to genome duplication (Tuskan *et al*, 2006), whereas LIM1 lacks such a closely related paralog (Fig. S1B). Thus, based on these criteria, we selected *LIM1* for further functional analysis.

"Altogether these results suggest that LIM1 is not a downstream target of SVL or FT1 pathway in LT response and distinct pathways MAY BE INVOLVED IN regulating repression of SVL and induction of LIM1 by LT."

Re: Thanks for indicating this. We have now changed this as suggested.

The data of Figure S4 could be combined to Figure 2.

Re: This has now been done.

Dear Rishi,

Thank you for submitting a revised version of your manuscript. We sent the manuscript to both of the original reviewers and have now received a report from one of them, which I have included below. As you will see, you have addressed the concerns satisfactorily. I have decided we can proceed towards publication, contingent on no overriding technical concerns being raised at this late stage by Reviewer #1. Before I can finally accept the manuscript, there are some remaining editorial points which need to be addressed. In this regard would you please:

- change Emmanuelle's middle initial to 'M' in our online submission system,
- acknowledge funding from Marie Skłodowska-Curie Individual Fellowships (DECORE: SEP-210709191); the "Seed Grant" money under Dev. Scheme (6031) of Institutions of Eminence (IoE) Scheme of Banaras Hindu University; the French government in the framework of the IdEX Bordeaux University "Investments for the Future" program / GPR Bordeaux Plant Sciences; Leverhulme grant no. RPG-2019-267; the China Scholarship Council under the Grant CSC NO. 202206320357; and Shandong Provincial Natural Science Foundation (ZR2021QC058) our online submission system,
- include up to five keywords,
- include a data availability section which states: "This study includes no data deposited in external repositories.",
- remove the author credit section from the manuscript,
- complete the general information table checklist,
- upload Appendix file in PDF format,
- include a table of contents with page numbers should be placed on the title page of Appendix 1,
- include extra EV figures in the Appendix PDF (or main manuscript) with the nomenclature Appendix Figure S1-Sx with the corresponding callouts; the nomenclature for tables should be Appendix Table S1-Sx,
- save Source Data files as one figure/folder and then upload as .zip files. I.e. all the Source data files for figure 1 need to be saved in a single folder and this needs to be zipped and then uploaded as "SD figure 1.zip" file,
- provide legends for figures 2b-e are not provided in a sequential manner,
- provide figure titles for supplementary figures 1-9 in the manuscript,
- provide exact p values in the legends of figures 1a; 2b-c; 3a-d; 4a-b, d; 5a-d; 6c, and supplementary figure 2a,
- define the nature of n in the legends of supplementary figures 3a-b; 7a; 9a-b, and
- define error bars are not defined in the legends of supplementary figures 3a-b; 7a; 9a-b.

I am looking forward to receiving your revised manuscript.

EMBO Press is an editorially independent publishing platform for the development of EMBO scientific publications.

Best wishes,

William

William Teale, PhD
Editor
The EMBO Journal
w.teale@embojournal.org

We realize that it is difficult to revise to a specific deadline. In the interest of protecting the conceptual advance provided by the work, we recommend a revision within 3 months (7th Nov 2024). Please discuss the revision progress ahead of this time with the editor if you require more time to complete the revisions. Use the link below to submit your revision:

Referee #2:

My previous round of review comments have largely been addressed in this revised manuscript.

First, I would suggest the authors to integrate the additional data into the manuscript, including expression levels of GA20ox2 homologous genes.

Moreover, how many repeats for data of bud break rate (%) in all related figures? Looks like they lack SE or SD?

Dear William,

Please find enclosed our revised version in which we have addressed the editorial revisions and also included the supplementary figure for GA20-oxidase expression. All the changes are indicated in the revised manuscript and also in the point by point response indicated in response letter. Only editorial comment that we could not address was changing Emmanuelle middle name in online system. She is currently on vacation and she will be back in first week of September and therefore, if possible EMBO staff can affect this change as we are unable to do this. We hope the revisions are satisfactory.

Best wishes,

Response to Editorial and reviewer comments

Response to editorial comments:

- change Emmanuelle's middle initial to 'M' in our online submission system.

Re: Emmanuelle is currently on vacation and we are unable to change this online.

- acknowledge funding from Marie Skłodowska-Curie Individual Fellowships (DECORE: SEP-210709191); the "Seed Grant" money under Dev. Scheme (6031) of Institutions of Eminence (IoE) Scheme of Banaras Hindu University; the French government in the framework of the IdEX Bordeaux University "Investments for the Future" program / GPR Bordeaux Plant Sciences; Leverhulme grant no. RPG-2019-267; the China Scholarship Council under the Grant CSC NO. 202206320357; and Shandong Provincial Natural Science Foundation (ZR2021QC058) our online submission system.

Re: Edited in online submission system: Funding from Marie Skłodowska-Curie Individual Fellowships (DECORE: SEP-210709191); Wenner-gren fellowship (UPD2019-0203); Knut and Alice Wallenberg foundation grant (2023.0209); Vetenskapsradet (2020-03522); the "Seed Grant" money under Dev. Scheme (6031) of Institutions of Eminence (IoE) Scheme of Banaras Hindu University; the French government in the framework of the IdEX Bordeaux University "Investments for the Future" program / GPR Bordeaux Plant Sciences; Leverhulme grant no. RPG-2019-267; the China Scholarship Council under the Grant CSC NO. 202206320357; and Shandong Provincial Natural Science Foundation (ZR2021QC058).

- include up to five keywords.

Re: Included in the manuscript.

- include a data availability section which states: "This study includes no data deposited in external repositories."

Re: Included in the manuscript.

- remove the author credit section from the manuscript.

Re: Removed the author credit section from the manuscript.

- complete the general information table checklist.

Re: Completed.

- upload Appendix file in PDF format. include a table of contents with page numbers should be placed on the title page of Appendix 1.

Re: Included in the manuscript as indicated.

- include extra EV figures in the Appendix PDF (or main manuscript) with the nomenclature Appendix Figure S1-Sx with the corresponding callouts; the nomenclature for tables should be Appendix Table S1-Sx.

Re: Included in the manuscript as indicated.

- save Source Data files as one figure/folder and then upload as .zip files. I.e. all the Source data files for figure 1 need to be saved in a single folder and this needs to be zipped and then uploaded as "SD figure 1.zip" file.

Re: Arranged as indicated.

- provide legends for figures 2b-e are not provided in a sequential manner.

Re: Provided legend in sequential manner.

- provide figure titles for supplementary figures 1-9 in the manuscript.

Re: Provided figure titles for supplementary figures.

- provide exact p values in the legends of figures 1a; 2b-c; 3a-d; 4a-b, d; 5a-d; 6c, and supplementary figure 2a.

- Re: Provided p values for unpaired t-tests as indicated. However where we performed one way ANOVA for multiple comparisons, indicated significance is shown with different letters.

- define the nature of n in the legends of supplementary figures 3a-b; 7a; 9a-b and define error bars are not defined in the legends of supplementary figures 3a-b; 7a; 9a-b.

Re: defined the nature of n and error bars in the legends as indicated.

Response to Reviewer

My previous round of review comments have largely been addressed in this revised manuscript.

- First, I would suggest the authors to integrate the additional data into the manuscript, including expression levels of GA20ox2 homologous genes.

Re: Included additional data into the manuscript.

- Moreover, how many repeats for data of bud break rate (%) in all related figures? Looks like they lack SE or SD?

Re: Included information in the figure legends.

Dear Rishi,

I am pleased to inform you that your manuscript has been accepted for publication in the EMBO Journal.

Congratulations on a really interesting study!

Best wishes,

Will

William Teale, PhD
Editor
The EMBO Journal
w.teale@embojournal.org
